# A carbon nanotube tape for serial-section electron microscopy of brain ultrastructure

Yoshiyuki Kubota [1,2], Jaerin Sohn[1,3], Sayuri Hatada[1], Meike Schurr[4], Jakob Straehle[4], Anjali Gour[4], Ralph Neujahr[5], Takafumi Miki[6], Shawn Mikula[7] & Yasuo Kawaguchi [1,2]

Automated tape-collecting ultramicrotomy in conjunction with scanning electron microscopy (SEM) is a powerful approach for volume electron microscopy and three-dimensional neuronal circuit analysis. Current tapes are limited by section wrinkle formation, surface scratches and sample charging during imaging. Here we show that a plasma-hydrophilized carbon nanotube (CNT)-coated polyethylene terephthalate (PET) tape effectively resolves these issues and produces SEM images of comparable quality to those from transmission electron microscopy. CNT tape can withstand multiple rounds of imaging, offer low surface resistance across the entire tape length and generate no wrinkles during the collection of ultrathin sections. When combined with an enhanced en bloc staining protocol, CNT tape-processed brain sections reveal detailed synaptic ultrastructure. In addition, CNT tape is compatible with post-embedding immunostaining for light and electron microscopy. We conclude that CNT tape can enable high-resolution volume electron microscopy for brain ultrastructure analysis.

[1] Division of Cerebral Circuitry, National Institute for Physiological Sciences, 5-1 Myodaiji-Higashiyama, Okazaki, Aichi 444-8787, Japan. [2] Department of Physiological Sciences, The Graduate University for Advanced Studies (SOKENDAI), 5-1 Myodaiji-Higashiyama, Okazaki, Aichi 444-8787, Japan. [3] Research Fellow of Japan Society for the Promotion of Science (JSPS), 5-3-1 Kojimachi, Chiyoda-ku, Tokyo 102-0083, Japan. [4] Department of Connectomics, Max-Planck Institute for Brain Research, Max-von-Laue-Str. 4, D-60438 Frankfurt, Germany. [5] Carl Zeiss Microscopy GmbH, ZEISS Microscopy Customer Center Europe, Rudolph-Eber-Str. 2, D- 873447 Oberkochen, Germany. [6] Graduate School of Brain Science, Doshisha University, 1-3 Tatara Miyakodani, Kyotanabe, Kyoto 610-0394, Japan. [7] Electrons–Photons–Neurons, Max-Planck Institute of Neurobiology, Am Klopferspitz 18, D-82152 Martinsried, Germany. Correspondence and requests for materials should be addressed to Y.K. (email: yoshiy@nips.ac.jp)

The electron microscopy (EM)-based reconstruction of neuronal circuits from serial ultrathin sections has attracted considerable recent attention, despite the emergence of super resolution microscopy[1], because EM is a reliable method for the diverse-scale analysis of dense nanoscale details in biological structures. Such structures include the entire nervous system[2], retina[3,4], cortex[5–8], myelin sheaths[9], endoplasmic reticulum[10,11], renal pelvis[12], cornea[13], mitochondria[14], Drosophila brain[15,16], plant tissue[17] and viral proteins[18]. EM volume datasets are typically obtained using methods, such as focused ion beam-scanning electron microscopy (FIB-SEM)[8,19,20], serial block-face electron microscopy (SBEM)[4,21–23], automated tape-collecting ultramicrotomy (ATUM) with SEM[24–26], transmission electron microscope camera array (TEMCA)[7,27], and transmission-mode SEM[28] in addition to conventional EM using ultramicrotomes with transmission electron microscopy (TEM)[6,29,30], with each method possessing unique benefits and drawbacks[31].

Here we focus on the ATUM method, which allows for efficient, automated collection of thousands of serial ultrathin sections of uniform quality that subsequently can be imaged with SEM[24]. Currently, the most commonly used tape for ATUM is carbon coated (cc)-Kapton tape (polyimide film, DuPont, Wilmington, USA)[5,24,32], but it has deficiencies due to a relatively high sheet resistance, non-uniform carbon coating that causes mottled surface resistance and scratches. Moreover, there is no assurance of a regular supply of high-quality cc-Kapton tape due to inconsistent industrial production procedures. Therefore, an improved alternative tape would have a valuable role in the field of ATUM-based EM.

Until recently the analysis of synaptic connections was performed with serial images obtained using TEM[2,6,29,33–35] where images are captured using an electron beam transmitted through ultrathin sections with a high acceleration voltage of 80–300 keV. Tissue sections processed with a protocol involving an en bloc osmium staining procedure can produce sufficient contrast for TEM observation[6,29,35,36]. After the introduction of the large volume EM methods using SBEM[21] for automated serial image acquisition, the necessity of increasing heavy metal density in the tissue block became apparent under low acceleration voltages below 10 keV. For that reason, a modified staining protocol was developed[37] to improve heavy metal density in tissue sections including en bloc reduced osmium tetroxide-thiocarbohydrazide (TCH)-osmium (rOTO)[38], uranyl acetate and lead aspartate staining that could increase tissue conductivity and image contrast for higher imaging throughput. However, the increased heavy metal staining precludes observation of fine cellular membrane structures[38] such as synaptic clefts and modified en bloc heavy metal staining procedures are necessary to achieve well-preserved ultrastructure with high conductivity and sample contrast for fine-scale analysis.

In this study, we address these issues by introducing an improved tape and tissue staining protocol. We screened candidates and found that plasma-hydrophilized-carbon nanotube (CNT) tape is optimal due to its extremely high surface conductivity and low endogenous signal, and it can provide high-quality images of tissue sections with SEM. We also developed a modified staining protocol in which the TCH step was excluded from an earlier protocol[37] for resolving fine ultrastructure. Altogether, these methods will improve ATUM-based serial section imaging and facilitate brain microcircuit analysis.

## Results

### Limitations of cc-Kapton tape for ATUM-SEM.
We quantitatively evaluated the commonly used cc-Kapton tape and found that it did not consistently provide adequate imaging conditions for ATUM-SEM. For example, the surface sheet resistance was very high (19.2, 107, and 6,530 MΩ $\square^{-1}$; i.e., megaohms per square) for three samples of cc-Kapton tape rolls (Boeckeler Instruments, Inc., Tucson, AZ, USA) and the vacuum deposited carbon was not always uniformly coated on the Kapton tape, causing occasional charging problems during SEM imaging (Supplementary Fig. 1). Scratches were also frequently found on the surface, which produced substantial image artifacts (Supplementary Fig. 1). Considering these problems and the fact that usable cc-Kapton tape is not consistently supplied commercially we concluded that the cc-Kapton tape does not reliably provide good imaging conditions and prompted us to search for a more optimal tape for ATUM. We tested many different tapes in addition to the CNT tape including: copper foil, 8 mm videocassette, ITO (indium tin oxide) coated PET, germanium-coated Kapton, and open-reel (Supplementary Fig. 2, Supplementary Table 1), but none of these worked well by our screening criteria (see Supplementary Notes 1–3 for more details).

### Properties of CNT tape for ATUM.
The properties such as conductivity, hydrophilicity, resistance to beam damage, mechanical and chemical strength and surface structure are important for an ATUM tape. The CNT tape was found to be superior or comparable to the cc-Kapton tape in most of these properties (Supplementary Figs. 3-7, Supplementary Note 4). CNTs are a flexible carbon allotrope of cylindrical nanostructure that was discovered in the early 1990s[39]. We used a CNT tape composed of three layers: an overcoat layer with CNTs (about 2 μm thick), a PET layer (50 μm thick) and a hard coat layer (about 2 μm thick) (Fig. 1a) and found a very-low surface resistance uniformly on the tape surface (242 ± 18.0 Ω $\square^{-1}$, between 223–305 Ω $\square^{-1}$, over a 10-meter length) (Fig. 1b), orders-of-magnitude lower than that of the cc-Kapton tape (Boeckeler Instruments, Inc., Tucson, USA). We observed good resilience and tape handling was easy enough to adhere tape strips to a flat wafer surface. To obtain a good image that lacks charging artifacts, we put double-sided adhesive conductive tape (Nisshin EM Co., Ltd., Tokyo, Japan) on a 4-inch silicon wafer and adhered the CNT tape with sections on top. The CNT layer was grounded to the wafer with a copper foil tape to secure an escape route for incident electrons (Supplementary Fig. 2). It did not generate any noticeable noise with either an in-lens secondary electron (SE) detector (In-lens SE) or a backscattered electron (BSE) detector (BSD) using optimized imaging conditions. High-quality images of 50 nm-thick sections processed with a modified heavy metal staining (mHMS) protocol (Fig. 2a, b) comparable to images of sections on cc-Kapton tape (Fig. 2c, d) were obtained.

Despite its superior qualities, we identified two problems with the CNT tape: generation of wrinkles on collected ultrathin sections and depression damage caused by the beam during SEM imaging, which may compromise high-quality imaging.

### A plasma discharge pre-treatment prevents section wrinkles.
Plasma discharge pre-treatment of the CNT tape was assessed for whether it effectively blocks the generation of wrinkles on collected ultrathin sections using ATUM. Copious wrinkles, which can prevent the ability to trace neural circuits through successive serial ultrathin section images, were found in the ultrathin sections we collected (Fig. 1c). We hypothesized that the wrinkles were generated due to the hydrophobic nature of the tape surface. Indeed, the pure water contact angle (PWCA) of the CNT tape was 79.5 degrees (Fig. 1d). To hydrophilize the tape, we used a slit type atmospheric pressure plasma generator (A-1000, SAKIGAKE-Semiconductor Co., Ltd., Kyoto, Japan) and a

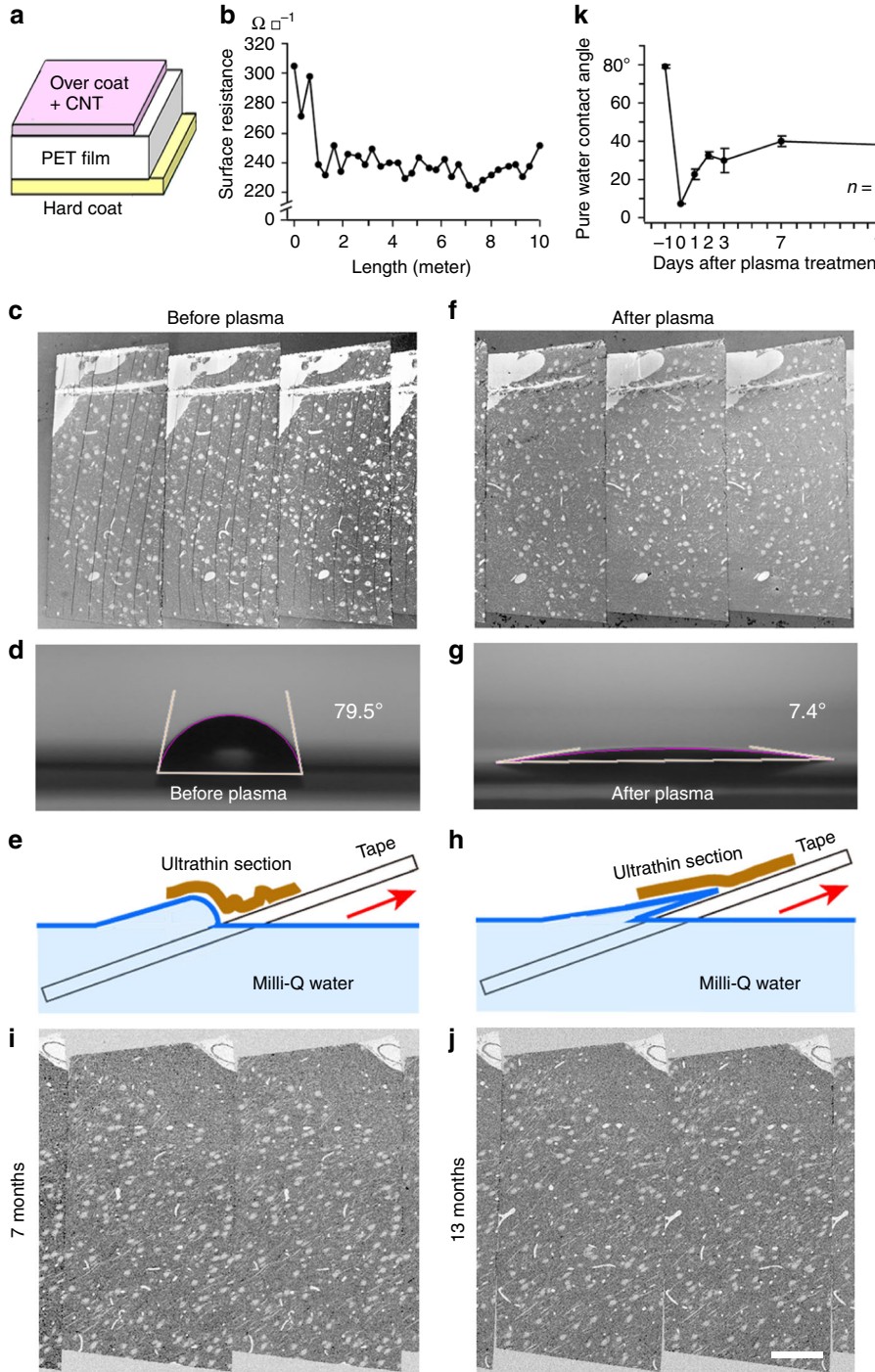

**Fig. 1** Plasma glow discharge treatment prevents the generation of copious wrinkles on the sections collected on CNT tape. **a** CNT-coated PET tape consists of three layers. The middle layer is a 50 μm-thick core structure made of PET film. CNTs are buried in the over coat layer (2 μm-thick). A hard coat layer (2 μm-thick) is on the opposing side of the PET film. Both coats are composed of a non-disclosed polymer. **b** Surface resistance of the CNT tape is uniform. **c** Ultrathin sections on the CNT-coated PET tape show many wrinkles. **d** PWCA of the CNT tape is 79.5 degrees. **e** Possible mechanism of the plasma treatment effect on the collection of the ultrathin sections for the untreated tape with the steep PWCA may cause difficulty in section landing. **f** Ultrathin sections on the CNT-coated PET tape with plasma treatment show no wrinkles. **g** PWCA of the CNT-coated PET tape after the plasma discharge treatment becomes 7.4 degrees, which shows that the tape is very hydrophilic. **h** The shallow PWCA promotes a smooth landing of the ultrathin sections on the plasma-treated tape from the water surface. **i, j** The plasma treatment effect for no wrinkle generation lasts for 7 months (**i**) and 13 months (**j**). Scale, 100 μm. **k** Time course of the plasma treatment effect on the CNT-coated tape indicated by PWCA. Error bars denote SD. Scale in **j** is for **c**, **f**, **i**

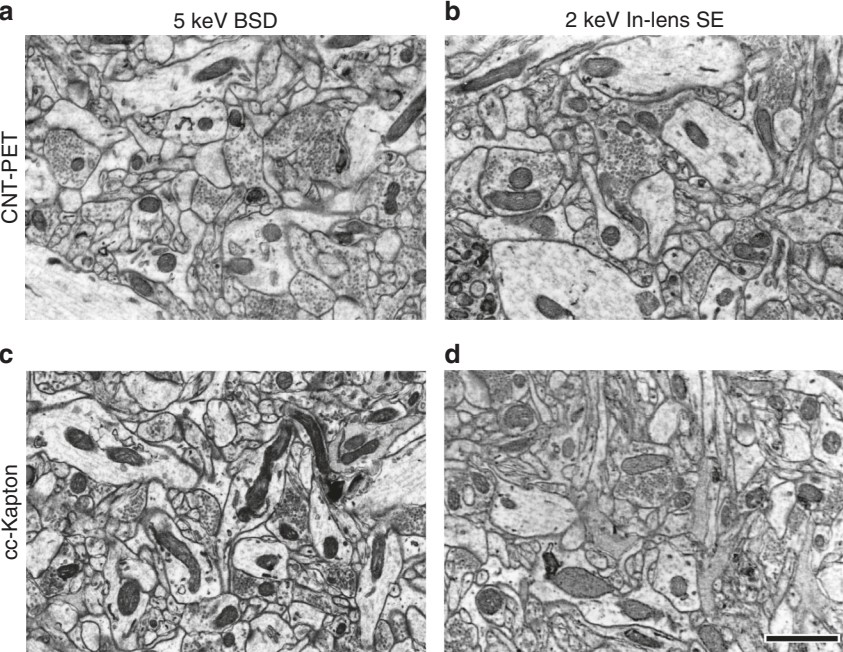

**Fig. 2** Images of ultrathin sections on CNT-PET tape and cc-Kapton tape are comparable. **a–d**, Images of mHMS-treated brain tissue captured with the BSD at 5 keV, 3.2 µs dwell time (**a**, **c**) or the In-lens SE detector at 2 keV, 3.2 µs dwell time (**b**, **d**) on CNT-coated PET tape (**a**, **b**) or on cc-Kapton tape (**c**, **d**). Scale in **d**, 1 µm, is also for **a–c**

custom-made reel-to-reel motorized winder (Supplementary Fig. 1).

The hydrophilization of the CNT tape by plasma discharge (1 cm s$^{-1}$) enabled us to collect ultrathin sections without any wrinkles (Fig. 1f). After treatment, the CNT tape had a shallow PWCA (7.4 degrees, Fig. 1g) that promoted the smooth deposition of ultrathin sections on the tape, as shown in the explanatory drawing (Fig. 1h). In contrast, the untreated hydrophobic tape surface may cause wrinkles due to the large water contact angle (Fig. 1d, e). The effect of the plasma treatment decreased over time, with an initial drop of the plasma effect (PWCA: 30°) by the second day, and then a plateau of hydrophilicity (PWCA: about 40°) (Fig. 1k). We kept the plasma-treated CNT tape in a desiccator and found that the plasma-treated CNT tape hydrophilization lasted up to 13 months (Fig. 1i, j), whereas plasma-treated cc-Kapton tape hydrophilization lasts only a few weeks (Pat Brey, personal communication). Moreover, the CNT tape may partially lose conductivity with plasma discharge treatment (12.6% reduction of conductivity; sheet resistance, $257 \pm 4.5$ and $294 \pm 18.1$ Ω □$^{-1}$ without and with plasma discharge treatment, respectively, $n = 3$ for each CNT tape; calculated conductance, 77.9 and 68.0 S m$^{-1}$, respectively), but the loss was less than with the cc-Kapton tape (19.3% reduction of conductivity, see Supplementary Note 1). The long-lasting plasma treatment effect on the CNT tape is advantageous for commercial and stock supply of ready-to-use tape.

**Depression damage has no negative effect on image quality**. We examined whether the depression damage might negatively influence repeated imaging using BSD by capturing 2048 × 2048 pixel images at a high magnification with 2.8 ke$^-$ nm$^{-2}$ electron dose, repeatedly (1, 5, 10, and 20 times, Fig. 3). We found that there was no striking degradation of the image even after imaging 20 times with the BSD at the same conditions with 5 keV with respect to beam damage (Fig. 3i). We also wondered if repeated imaging might negatively influence the stitching of image tiles in a large mosaic due to repeated beam exposure along overlapping

tile edges. We then captured a large rectangular image, including the four exposed domains captured with the BSD, In-lens SE and Everhart-Thornley detector (ETD) in SEM and 3D laser scanning confocal microscope (SCM) (VX-250, Keyence Corporation, Osaka, Japan) (Fig. 3a–d). Although we found a significant depression of the imaged domains, which was dependent on the number of imaging instances (Fig. 3c, d), gaps or warping were not found in the image captured with the BSD, nor with the In-lens SE detector (Fig. 3a, b, e–h). The imaging focus depth was sufficiently large so that the image from the depression area still remained clearly focused. We then verified that the depression damage caused by the electron beam did not cause subsequent stitching problems (Supplementary Fig. 8).

Imaging with the In-lens SE detector showed a substantial contaminated area likely as a result of a thin layer of adventitious carbon build up (Fig. 3b, g, h) from where the repeated images were obtained, although it was barely visible with the BSD (Fig. 3a, e, f). Less contamination was found in areas imaged by the In-lens SE detector, where the electron dose was about 20 % of the dose when using the BSD at the same acceleration voltage (Supplementary Figs. 4, 5). The contamination was hardly observable at 1.5 keV (Supplementary Fig. 5). The contaminated dark overlay on the image is believed to be due to the result of beam interaction with the epoxy resin, which resulted in contamination inside the chamber. We also found the same results in images obtained using cc-Kapton tape (data not shown). This is likely to be more visible in In-lens SE images than in BSE images, because the In-lens SE detects structures on the surface, for example contamination, more than the BSD. These results indicate that the depression for imaging under normal conditions may have very little influence on imaging and stitching of tiled images.

**CNT tape applications for light microscopy**. One significant feature of the ATUM is that it can be used for array tomography[40]. For light microscopy (LM) use, the tape must be transparent and emit no auto-fluorescence for an excitation

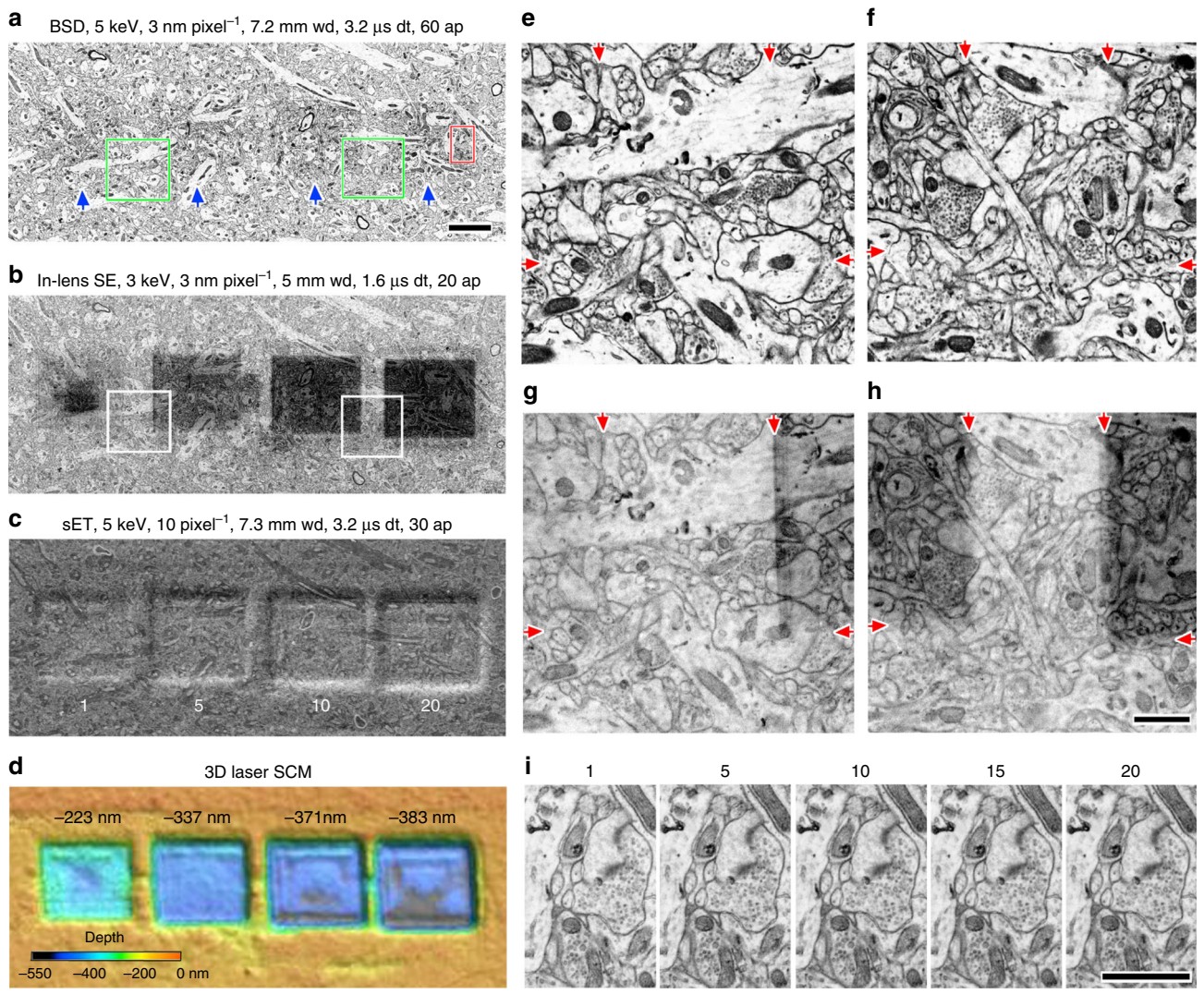

**Fig. 3** Repeated image capturing does not substantially affect image quality. **a** Image of the mHMS ultrathin section of cortex captured with BSD for 3.2 μs dwell time, 3 nm pixel$^{-1}$, 60 μm aperture at 5 keV, 2.8 ke$^-$ nm$^{-2}$ electron dose, where the 2048×2048 image size, 3.2 μs dwell time, 3 nm pixel$^{-1}$, 60 μm aperture at 5 keV had been captured one, 5, 10 and 20 times indicated by blue arrows from left to right. Scale, 3 μm, is also for **b**–**d**. **b** Image of the same location of **a** captured with an In-lens SE detector for 1.6 μs dwell time, 3 nm pixel$^{-1}$, 20 μm aperture at 3 keV. Darker squares correspond with the pre-imaged area. **c** The same area imaged with the ETD. The imaging times are indicated below the depression square. **d** The surface profile image. Depression depth is indicated above the depression square. **e**. Enlarged image of left green square in **a**. Arrows on the top left and middle left indicate the border of the once imaged area. Arrows on the top right and middle right indicate the border of the area imaged 5 times. **f** Enlarged image of right green square in **a**. Arrows on the top left and middle left indicate the border of the area imaged 10 times. Arrows on the top right and middle right indicate the border of the area imaged 20 times. **g** Enlarged image of left square in **b**. Arrows indicate the border shown in **e**. **h** Enlarged image of right square in **b**. Arrows indicate the border shown in **f**. Scale, 1 μm, is also for **e**–**g**. **i** Enlarged image of red rectangle in **a**. The number of imaging times are shown above the images. Scale, 1 μm. wd, working distance; dt, dwell time; ap, aperture

wavelength. The CNT tape is transparent (Supplementary Fig. 2) and has no auto-fluorescence when excited by yellow-green ($\lambda = 561$ nm) or red light ($\lambda = 633$ nm), only a very faint auto-fluorescence when excited with blue light ($\lambda = 488$ nm), and slightly more with purple light ($\lambda = 405$ nm) (Fig. 4a). Absorption was low overall (Fig. 4b), with a light transmittance of about 80% (Fig. 4c), total light transmittance of 88.4% (Fig. 4d) and haze of 1.9% (Fig. 4e). Therefore, the tape can be used for multi-color fluorescent immunohistochemistry or array tomography[40] except for excitation with violet light.

We examined serial ultrathin sections of brain tissue processed with a twice osmium protocol (TO) or twice osmium plus lead aspartate protocol (TOLA) with gamma-aminobutyric acid (GABA) post-embedding immunohistochemistry using a 15 nm colloidal gold-labeled secondary antiserum[29,33,35]. The GABA labeling was specific to inhibitory neuronal profiles (Fig. 4f–i, Supplementary Movie 1) and confirmed the capability of immunoreactions using CNT tape. We also examined serial thin sections with the immunofluorescent method and found GABA-positive fluorescent immunoreactivity (Fig. 4j, k). On the other hand, because cc-Kapton tape is yellowish-brown in color (Supplementary Fig. 2) and showed significant auto-fluorescence in all emission light wavelengths (Fig. 4a), the light transmittance was lower than the CNT tape (Fig. 4c) and the total light transmittance was 47.9% (Fig. 4d). We performed GABA post-embedding immunoreactions on ultrathin sections on cc-Kapton tape and found no immunoreactivity, with many non-specific gold particles on the surface of the cc-Kapton tape. Probably the carbon on the cc-Kapton tape restored adsorptive capacity by thermal reactivation during the carbon evaporation

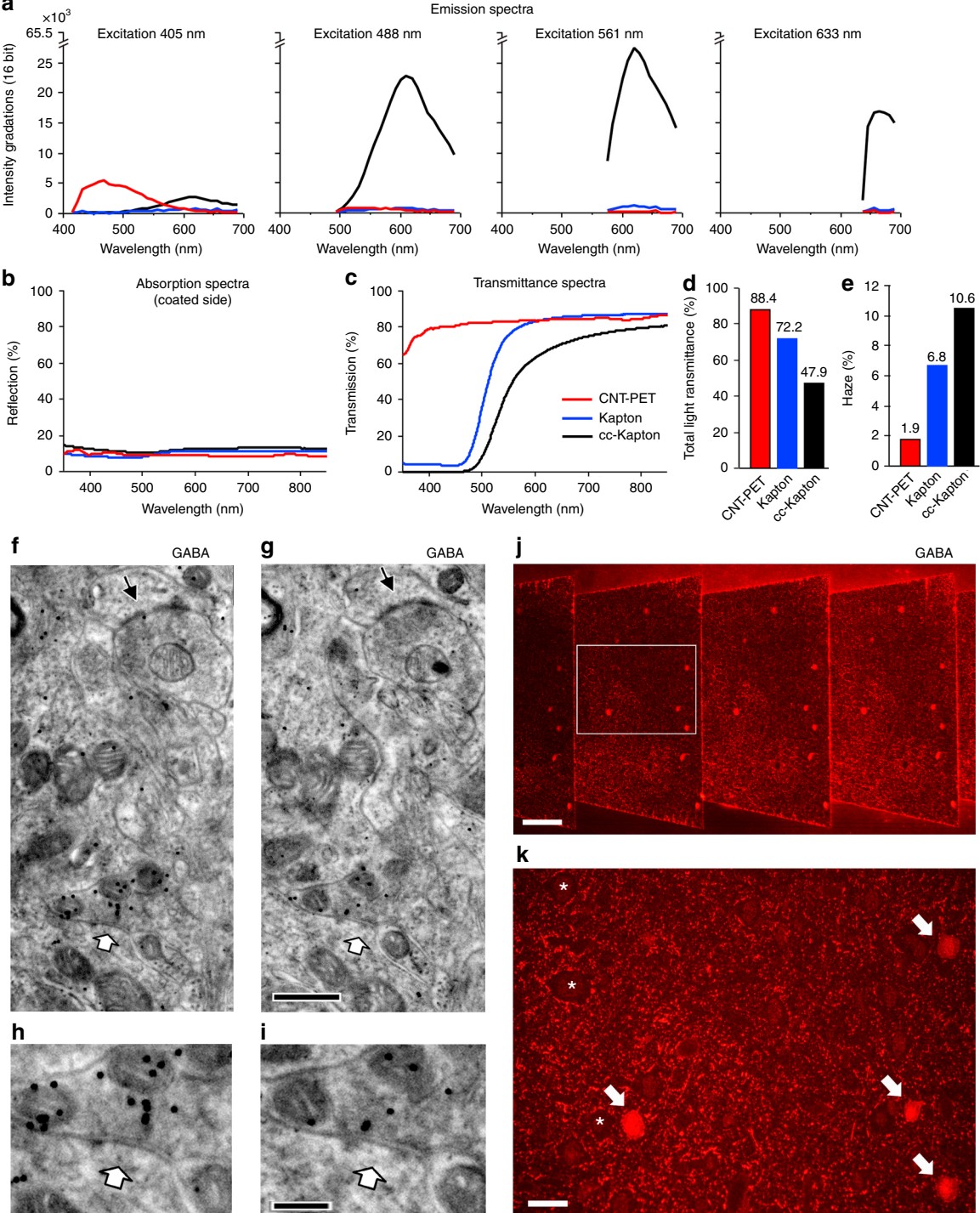

**Fig. 4** CNT tape is feasible for light microscopy. **a** Emission spectrum, **b** Absorption spectrum, **c** Transmission spectrum, **d** Total light transmission, and **e** Haze of the CNT, uncoated Kapton and cc-Kapton tape. **f**, **g** GABA post-embedding immunoreaction with 15 nm colloidal gold particles on serial sections of rat striatum processed with TO on the CNT tape imaged with BSD at 5 keV, 3.2 μs dwell time. GABAergic synapse (white arrow) and non-GABAergic synapse (arrow). Scale, 0.5 μm. **h**, **i** Enlarged image of GABAergic synaptic contact indicated by a white arrow in **f** and **g**. Scale, 0.2 μm. **j** GABA post-embedding immunofluorescent reaction on serial thin sections of rat cortex processed with TOLA on the CNT tape. Scale, 80 μm. **k** Enlarged image of rectangle in **j**. GABA immunoreactive neurons (arrows) and basket like boutons (asterisks). Scale, 20 μm

and adsorbed the primary and secondary antisera, which prevented GABA immunoreactivity. These results showed that the CNT tape, but not the cc-Kapton tape, is useful for fluorescence microscopy, suggesting that the CNT tape compares favorably with Kapton tape for use in ATUM.

**A modified tissue staining protocol for high-resolution imaging**. The high conductivity of CNT tape is advantageous because it permits imaging using high electron doses (11 ke⁻ nm⁻² or more). As a result, high-resolution image acquisition of synaptic structures is possible. Due to the high conductivity of the tape, we

**Table 1 Detailed comparison of staining steps in employed staining protocols**

| Protocol | BROPA (Mikula 2015) | Hua (2015) | mHMS | TO | TOLA |
|---|---|---|---|---|---|
| Tissue size | Whole mouse brain | 1 mm φ × 2 mm block | 50 µm-thick section | 50 µm-thick section | 50 µm-thick section |
| Step 1 | 40 mM $OsO_4$, 35 mM $K_4[Fe(CN)_6]$, 2.5 M formamide, 0.1 M cacodylate, pH 7.4 | 2% $OsO_4$, 0.15 M cacodylate, pH 7.4 | 2% $OsO_4$, 1.5% potassium ferrocyanide, 0.1 M PB, pH 7.4 | 1% $OsO_4$, 1.5% potassium ferrocyanide, 0.1 M PB, pH 7.4 | 1% or 2% $OsO_4$, 1.5% potassium ferrocyanide, 0.1 M PB, pH 7.4 |
| Time @ temp. 1 | 96 h @ RT | 1.5h @ RT | 1h @ 4 °C | 1h @ 4 °C | 1h @ 4 °C |
| Wash 1 | ⇓ | ⇓ | 10 min wash in water ×3 | 10 min wash in 0.1 M PB ×3 | 10 min wash in 0.1 M PB ×3 |
| Step 2 | 40 mM $OsO_4$, 0.1 M cacodylate, pH 7.4 | 2.5% potassium ferrocyanide, 0.15 M cacodylate, pH 7.4 | ⇓ | ⇓ | ⇓ |
| Time @ temp. 2 | 96 h @ RT | 1.5h @ RT | ⇓ | ⇓ | ⇓ |
| Wash 2 | 4 h wash in 0.1 M cacodylate | 30 min wash in water ×2 | ⇓ | ⇓ | ⇓ |
| Step 3 | 320 mM pyrogallol unbuffered, pH 4.1 | 1% TCH unbuffered | 1% TCH unbuffered | ⇓ | ⇓ |
| Time @ temp. 3 | 72 h @ RT | 45 min @ RT | 20 min @ RT | ⇓ | ⇓ |
| Wash 3 | 4 h wash in 0.1 M cacodylate | 30 min wash in water ×2 | 10 min wash in water ×3 | ⇓ | ⇓ |
| Step 4 | 40 mM $OsO_4$ unbuffered | 2% $OsO_4$ unbuffered | 2% $OsO_4$ unbuffered | 1% $OsO_4$, 0.1 M PB, pH 7.4 | 1% or 2% $OsO_4$, 0.1 M PB, pH 7.4 |
| Time @ temp. 4 | 96h @ RT | 1.5h @ RT | 0.5h @ RT | 1h @ RT | 1h @ RT |
| Wash 4 | 4h wash in water | 30 min wash in water ×2 | 10 min wash in water ×3 | 10 min wash in water ×3 | 10 min wash in water ×3 |
| Step 5 | ⇓ | 1% uranyl acetate unbuffered | 1% uranyl acetate unbuffered | 1% uranyl acetate unbuffered | 1% uranyl acetate unbuffered |
| Time @ temp. 5 | ⇓ | overnight @ 4 °C 2h @ 50 °C | overnight @ RT | 40 min @ RT | overnight @ RT |
| Wash 5 | ⇓ | 30 min wash in water ×2 | 10 min wash in water x3 | 10 min wash in water x3 | 10 min wash in water x3 |
| Step 6 | ⇓ | Lead aspartate, pH 5.0 | Lead aspartate, pH 5.5 | ⇓ | Lead aspartate, pH 5.5 |
| Time @ temp. 6 | ⇓ | 2h @ 50 °C | 30 min @ 60 °C | | 1h @ 60 °C |
| Wash 6 | ⇓ | 30 min wash in water ×2 | 10 min wash in water ×3 | 10 min wash in water ×3 | 10 min wash in water ×3 |
| Step 7 | Dehydration, infiltration, embedding and hardening | Dehydration, infiltration, embedding and hardening | Dehydration, infiltration, embedding and hardening | Dehydration, infiltration, embedding and hardening | Dehydration, infiltration, embedding and hardening |
| Figure panels | Fig. 7b–e | Figs. 7f, g Sup Fig. 6f, 11, 12 | Figs. 1–3, 5a, g, m, s, Sup Fig. 1, 2, 4, 5, 6d, e, 7a–d, 8, 9 | Figs. 4f–k, 6b–d, h–j, n–p, t–v, x | Figs. 5e, f, k. l. q, r, w, y, 6, Sup Fig. 10 |

thought we could reduce the metal density from the more conventional mHMS procedure (Table 1) to achieve well-preserved ultrastructural detail suitable for the analysis of synapses with SEM. First, we tested how metal staining affects ultrastructure. Strong heavy metal staining protocols were developed for SBEM[3,22,37,41,42] which provide a metal sample of sufficient stain intensity to conduct electrons to ground, preventing charge accumulation and allowing for high-contrast images to be collected quickly. The procedure, mHMS (Table 1), is based upon a combination of the rOTO protocol[38], uranyl acetate and lead aspartate staining[37,42].

Despite remarkably high-contrast images, postsynaptic densities (PSD) were more lightly stained by this method than with conventional tissue processing for TEM by TO (Fig. 5a–d, g–j, Table 1). Identification of synaptic clefts and PSDs is important for detecting synaptic contacts[36,43] and may require observation at exactly the right angle or at a low acceleration voltage to collect information from a sufficiently small interaction volume (Supplementary Fig. 9). While it is generally difficult to identify symmetrical synapses and fine synaptic structures such as junction areas or small vesicle docking, asymmetrical synapses can be readily identified based on contact densities and vesicle clouds with the mHMS-treated tissue, where expanded heavy metal staining[38] may hide the synaptic cleft structure (Fig. 5a, g).

TCH, which bridges reduced osmium and osmium, was not used in our staining process to obtain quality images of fine ultrastructure. We made samples processed with TO or TOLA (Table 1). TO is a conventional histological method used for TEM samples, and TOLA is a modified mHMS protocol excluding the TCH step. Since TO treated tissue does not show enough contrast for SEM observation (Fig. 5b, h), we post-stained ultrathin sections with uranyl acetate/lead citrate to increase the metal content (Fig. 5c, d, i, j). Both of the protocols clearly improved image contrast. The cleft structures were shown clearly and PSDs showed heavier staining than the other cell membranes (Fig. 5c, d, i, j). The lead staining process for all serial sections on the tape

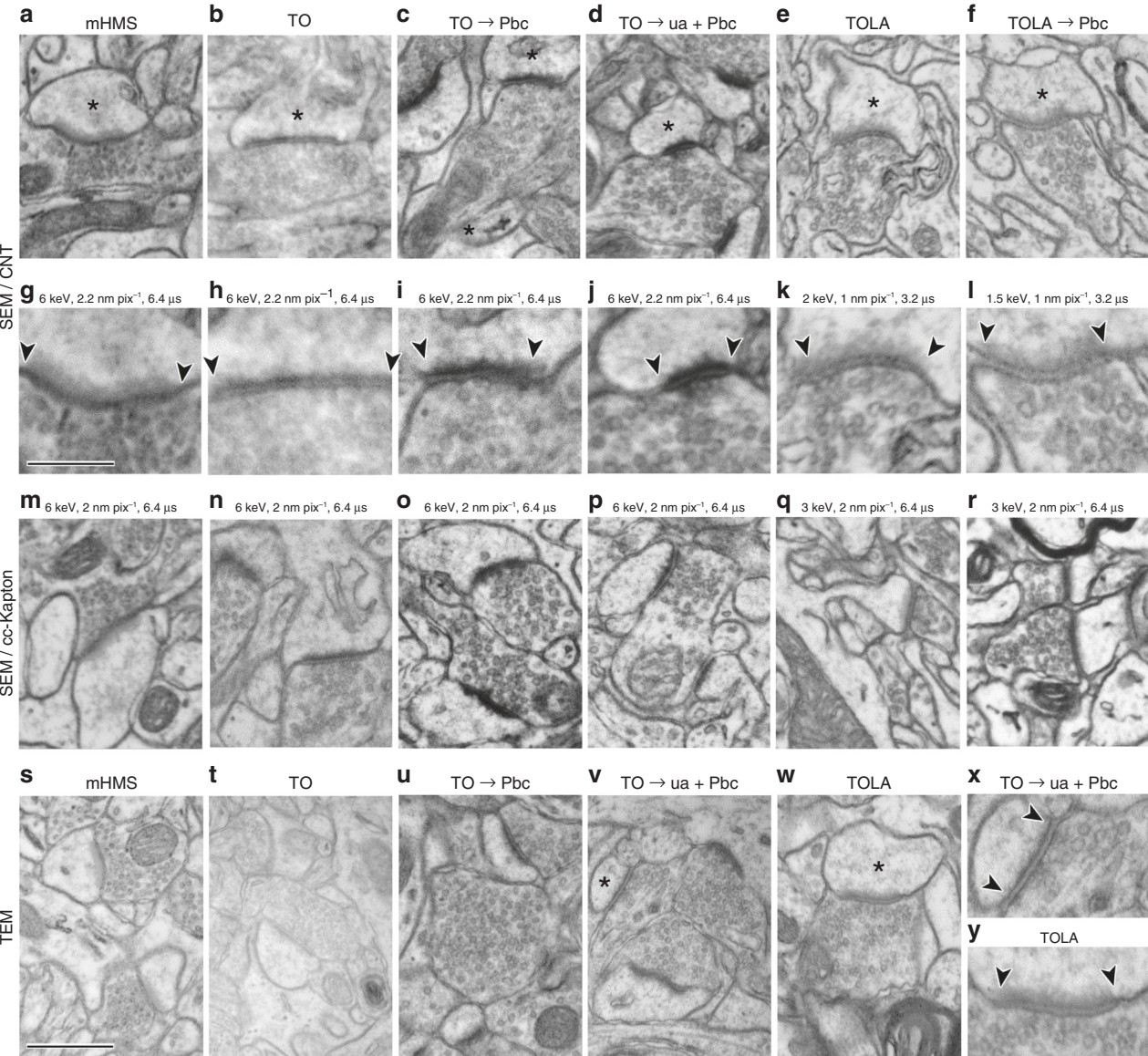

**Fig. 5** Different metal staining protocols make a difference in image quality of ultrathin sections, but images obtained with SEM or TEM are comparable. **a–l** Images of ultrathin sections of **a** mHMS, **b** TO, **c** TO with lead citrate (Pbc) section staining, **d** TO with uranyl acetate (ua) and lead citrate section staining, **e** TOLA, and **f** TOLA with lead citrate section staining on CNT-coated PET tape captured with a BSD. **g–l** Enlarged image showing synaptic junction in **a–f**, where postsynaptic spine heads are marked with an asterisk. Synaptic junction area is indicated with arrowheads. Scale in **g**, 0.25 µm, is for **h–l**, **x**, **y**. **m–r** Images of ultrathin sections of **m** mHMS, **n** TO, **o** TO with lead citrate section staining, **p** TO with uranyl acetate and lead citrate section staining, and **q** TOLA, and **r** TOLA with lead citrate section staining on cc-Kapton tape captured with a BSD. **s–w** Images of ultrathin sections of **s** mHMS, **t** TO, **u** TO with lead citrate section staining, **v** TO with uranyl acetate and lead citrate section staining, and **w** TOLA captured with TEM. Postsynaptic spine heads in **v** and **w** are marked with an asterisk. (**x** and **y**) Enlarged image showing synaptic junction of **v** and **w**. The synaptic junction area is marked with arrows. Scale in **s**, 0.5 µm, is for **a–f**, **m–w**

is laborious work. We added lead aspartate en bloc staining after TO staining, which is far easier than staining the sections separately, in order to increase the metal content in the tissue block.

The sections were processed with TOLA (Table 1), which provides sufficient image contrast without further staining and imaged with a BSD optimized with a low acceleration voltage (OnPoint BSE detector, Gatan Inc., Pleasanton, CA, U.S.A.) in an SEM (GeminiSEM 300, Carl Zeiss Microscopy GmbH, Oberkochen, Germany). The sections showed sufficiently high contrast (Fig. 5e, k). With lead citrate staining, the quality of the images was slightly improved, showing the lipid bilayer structure dimly (Fig. 5f, l). At 1.5 keV, the BSEs showed signal efficiently collected

only from the shallow depth of the brain tissue within the section. No background signal from the CNT tape below the section was observed for the BSE produced up to 53 nm into the tissue (Supplementary Fig. 9, Supplementary Table 2). The signal efficiency for a 50 nm thick sections was 99.5%. At 2 keV, the BSE's collected signals mostly from the 50 nm-thick tissue section and only faint background signals from the CNT tape. The signal efficiency for a 50 nm thick section was 83.0% (Supplementary Fig. 9, Supplementary Table 2) for high-contrast and resolution images (Figs. 5e, f, k, l and 6a, b, d, e, Supplementary Fig. 10), with almost no beam damage depression (Supplementary Fig. 4, 5). We concluded that ATUM preparations would be better with samples processed with the TOLA protocol, imaged using low acceleration

voltages and with the BSD optimized for low acceleration voltage detection.

Sections from the same tissue blocks were collected on cc-Kapton tape and imaged under similar conditions to compare the image quality with data collected using CNT tape. We found the images from the sections on the different tapes were comparable (Fig. 5). Finally, we wondered whether the SEM images are comparable to images obtained using TEM. The images captured by SEM were compared with those by TEM from the same tissue blocks with the different histological treatments: mHMS, TO and TOLA (Fig. 5). We found that the quality of the images captured with SEM was comparable to those collected with TEM. This indicates that the differences in ultrastructural properties were caused by the section staining procedure and not by the type of EM used (Fig. 5). Section processing with the TOLA protocol provided high-quality, high-resolution SEM images that were suitable for observation of synapse structure and comparable to conventional TEM images (Fig. 5).

Then we investigated how the acceleration voltage affects image quality using 50 nm TOLA processed sections. We captured images of the same region repeatedly using varied acceleration voltage strengths with the BSD detector (Fig. 6a) and with the In-lens SE detector (Fig. 6d) (Sigma, Carl Zeiss Microscopy GmbH, Oberkochen, Germany). We tried to collect optimal images at each acceleration voltage. We observed that imaging with 3–4 keV using a BSD (Fig. 6b), and with 1.5–2 keV using an In-lens SE (Fig. 6e) provided better images than those collected with other acceleration voltages. We quantitatively analyzed the images with varying acceleration voltages for image contrast using an intensity histogram[61]. (Fig. 6c, f). We found better values in both the Michelson contrast calculated at half width of the histogram and contrast-to-noise ratio (CNR)[44,45] for images captured with 3–4 keV using the BSD, and with 1.5–2 keV using the In-lens SE (Fig. 6g, Supplementary Table 3). These results support our subjective evaluation of image quality. The higher the acceleration voltage, the greater the background noise from the CNT tape, and, as a result, image contrast deteriorates. The lower the acceleration voltage, the lower the image contrast, due to reduced signal. Images of synaptic contacts looked better with 3–4 keV using the BSD, and with 1.5–2 keV using the In-lens SE (Fig. 6b, d). This relationship can be roughly estimated by the Monte Carlo simulation of electron trajectory in solids (CASINO)[46,47] (Supplementary Fig. 9), where the BSE interaction volume may correspond with the image signal generation site (read Supplementary Note 3 for more details). It can also depend to a large degree on tissue preparation and the performance of the SEM and detector. These parameters should be optimized for individual conditions and it is essential for users to investigate the optimized acceleration voltage for imaging.

**Practical applications of CNT tape for imaging**. It is important to know the applicability of CNT tape for different experimental projects. Therefore, we examined whether CNT tape can be used under different conditions in terms of type of tissue, metal composition of the block, imaging method, detector and SEM. We took serial images (QuantaFEG 200, FEI, Hillsboro, Oregon, U.S.A.) of coronal sections through the olfactory bulbs of a whole mouse brain prepared with the BROPA protocol[22] (Table 1) using a custom silicon-diode BSD (AXUV, International Radiation Detectors) and generated an image mosaic at 20 nm pixel$^{-1}$ (Fig. 7a–c). We found that the entire sections of the olfactory bulb collected onto the CNT tape were largely wrinkle-free and showed no charge accumulation (Fig. 7c). The high-magnification images were sufficient to observe the details of the ultrastructure of the neuronal profiles, including synaptic contacts (Fig. 7d, e). The CNT tape allowed for high-quality whole mouse brain imaging.

To test the applicability of CNT tape for high-throughput EM for connectomics, SE images of 35 nm sections of mouse primary somatosensory cortex prepared according to the Hua protocol[41] (Table 1) were taken with a 61 beam MultiSEM (MultiSEM 505, Carl Zeiss Microscopy GmbH, Oberkochen, Germany; Fig. 7f, g). Imaging with a MultiSEM requires that the conductive properties of the tape largely exceed those in a single beam SEM setup (~270 pA: using In-lens SE, 20 μm aperture size, 3 nm pixel$^{-1}$, ~6 keV with Sigma, Carl Zeiss Microscopy GmbH, Oberkochen, Germany) due to the higher total beam current (61 beam version, ~570 pA per beam; ~35 nA total). We found that images from sections on CNT tape showed no signs of charging and good quality images were acquired across a range of landing energies and pixel dwell times (Supplementary Fig. 11). We examined the impact of multiple exposures on image quality by imaging the same region 30 times. Up to 30 single exposures (landing energy: 1.5 keV, dose: 22.2 e$^-$ nm$^{-2}$ per beam, 667 e$^-$ nm$^{-2}$ total) were possible without considerable loss of image quality (Supplementary Fig. 12). These findings indicate that CNT tape is well suited to support high-throughput imaging with MultiSEM for sufficiently thick sections (≥30–35 nm) that block the signal of the underlying tape structure (Supplementary Fig. 6), which provides a sufficient resolution for the reconstruction of neuronal circuits[5,48].

## Discussion

In this study, we characterized a CNT tape for the ATUM-SEM system. Positive features include extremely high conductivity, good resilience and flexibility, low background signal, chemical and mechanical strength, vacuum compatibility and radiation-resistance comparable or superior to cc-Kapton tape. Images from sections on CNT tape were comparable to those from cc-Kapton tape or those obtained with a TEM. Image tiles could be seamlessly stitched for mosaics. Immunohistochemistry was possible for thin sections on CNT tape, but not on cc-Kapton tape. We also modified the mHMS protocol to obtain well-preserved ultrastructure suitable for the fine-scale analysis of synapses. Thus, CNT tape offers improvements in the investigation of microcircuits using large volume EM datasets.

In SEM, an electron's trajectory is determined by two main factors: the metal content of sections and the SEM beam acceleration voltage/landing energy. To acquire a high-resolution image, it is advantageous to increase image contrast with heavy metal staining, increase dwell time, decrease landing energy to reduce the interaction volume, and improve the detector efficiency, especially for low landing energies. The best conditions depend on the tissue preparation, detector and SEM type. Present results suggest that the highest image quality of the ATUM-SEM preparation can be obtained with low landing energies to reduce tape noise, a BSD optimized for low acceleration voltages, and a CNT tape. There is sufficient metal content in the tissue for SEM imaging, as images are captured within small pixel dwell times, and signals are collected from a small interaction volume within a shallow depth. Collectively, all of these changes result in an improved axial resolution and better signal-to-noise ratio[49,50].

An appropriate tape for ATUM must be conductive, hydrophilic, and flat, with good beam resistance, as well as resistance to physical or chemical influences, evenly coated in a conductive material, and yield low or no endogenous signal, including auto-fluorescence. Given its high conductive properties, the potential for multiple rounds of imaging, the smoothness of the surface, the long-lasting plasma discharge effect and the potential for post-embedding immunostaining and fluorescent light microscopy, we believe that CNT tape is an ideal substrate for large-scale connectomics projects. We also found that the degree of heavy metal

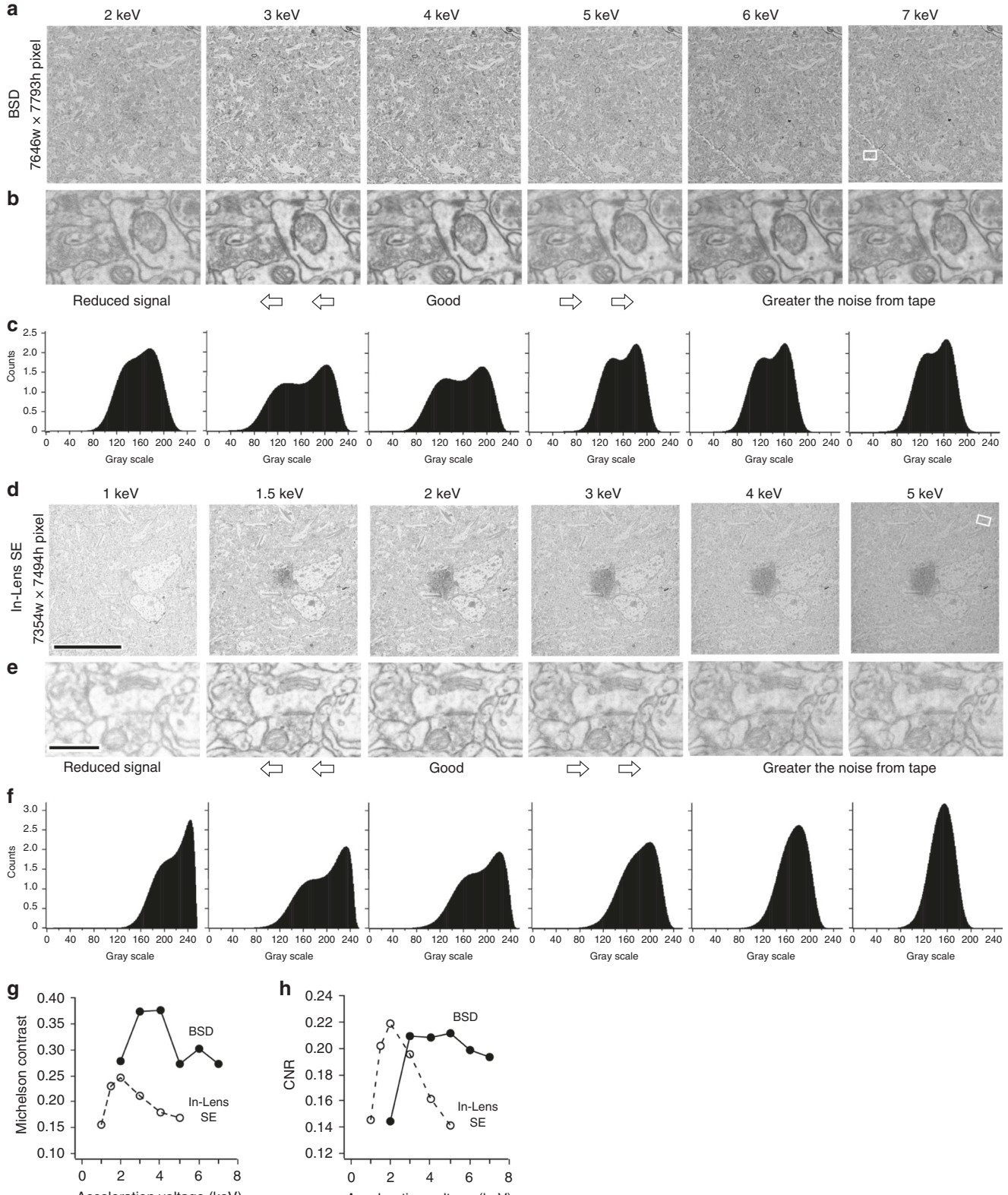

**Fig. 6** Image quality varies with different acceleration voltages. **a** Images of TOLA plus lead section stained brain tissue on the CNT tape are captured with a BSD for 3 nm pixel$^{-1}$, 7646 × 7793 pixel image size, 3.2 µs dwell time, 60 µm aperture, with different acceleration voltage strengths (2–7 keV) with optimized working distance (7.6 mm: 6 keV, 7.7 mm: 5 keV and 4 keV, 7.8 mm: 3 keV, 7.9 mm: 2 keV). **b** Enlarged images of each acceleration voltage showing a synapse located in the rectangle in right panel of **a**. **c** Intensity histogram of the image in **a**. **d** Images of the same brain tissue section are captured with an In-lens SE detector for 3 nm pixel$^{-1}$, 7354 × 7494 pixels image, 20 µm aperture, with different acceleration voltage strengths (1–5 keV) with optimized working distance (4.0 mm: 4–6 keV, 4.1 mm: 1–3 keV) on the CNT tape. Scale, 10 µm, is also for **a**. **e** Enlarged images of each acceleration voltage showing a synapse located in the rectangle in right panel of **d**. Scale, 0.5 µm, is also for **b**. **f** Intensity histogram of the image in **d**. **g** Michelson contrast values at half width of the intensity histogram. **h** Contrast-to-noise ratio (CNR) extracted from the intensity histogram

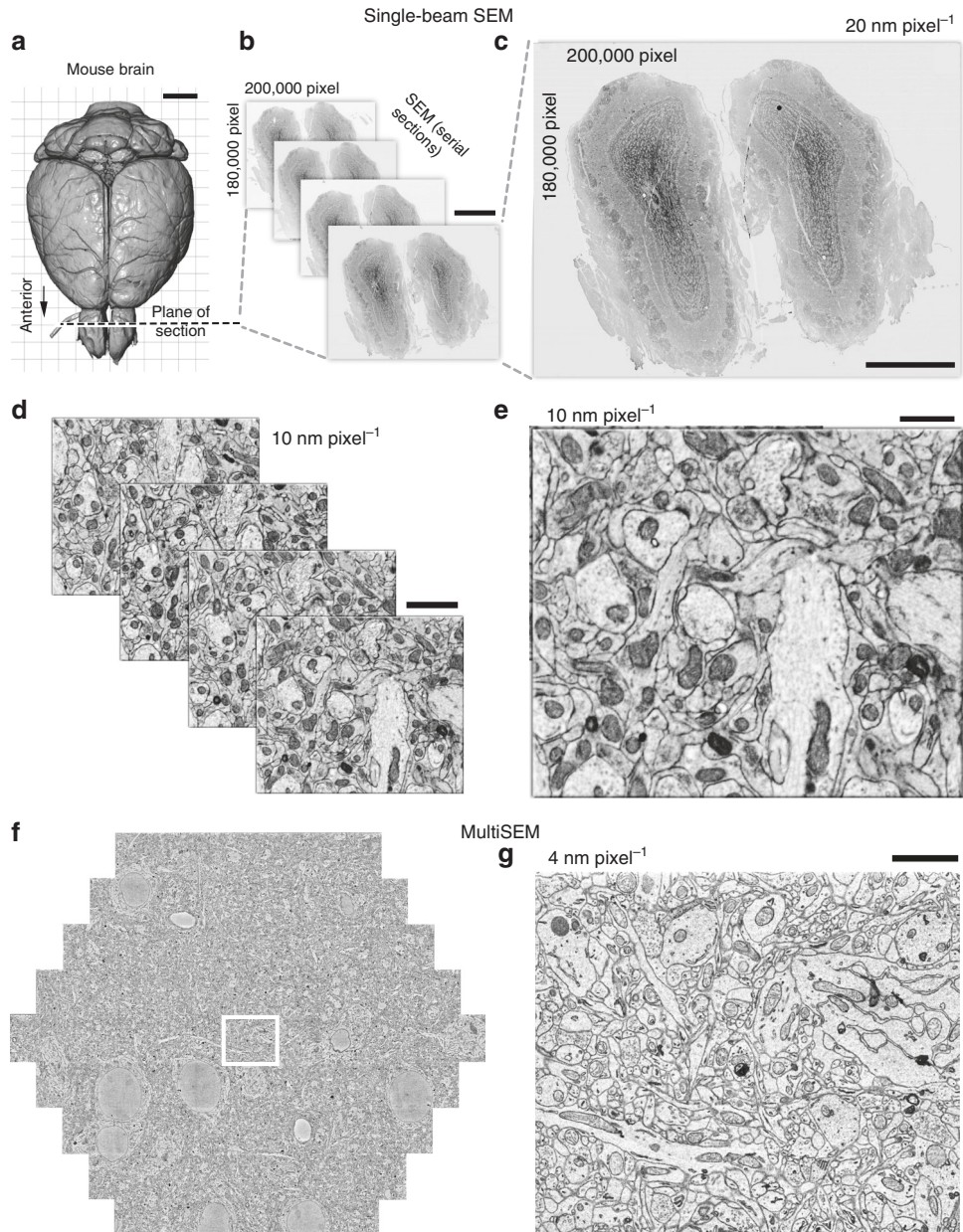

**Fig. 7** CNT tape provides good imaging conditions for large brain sections with single beam SEM (**a-e**) and MultiSEM (**f**, **g**). **a** 3D reconstruction of an adult mouse brain based on X-ray microCT. Scale, 2 mm. **b** Olfactory bulb serial sections (100 nm-thick) collected onto CNT tape are largely wrinkle-free and can be imaged with a BSE detector in high vacuum without charging. Scale, 1 mm. **c** Enlargement of an SEM mosaic, consisting of 3078 individual image tiles, with areas of the olfactory bulb indicated. Scale, 1 mm. **d** High-magnification stacks through the serial sections shows intact cellular membranes and well-stained synapses suitable for neuronal circuit reconstructions. Scale, 2 μm. **e** Enlarged serial section from **d**. Scale, 1 μm. **f** MultiSEM image of mouse primary somatosensory cortex captured with 4 nm pixel$^{-1}$, 3128 × 2724 pixels image size for each beam, 100 ns dwell time and 1.5 keV landing energy. Stitched hexagon of ~35 nm thick section of cortical mouse brain tissue. Scale, 10 μm. **g** Enlarged single tile image outlined by the white rectangle in **f**. Scale, 2 μm

staining in the brain tissue greatly affects visible ultrastructure. The stronger metal staining protocol, mHMS, stains the tissue too densely, resulting in images that make the identification of inhibitory synapses difficult, but with contrast adequate for detecting most excitatory synapses. An advantage of heavy metal staining is that the electron beam can scan samples quickly, that is, in a few hundred nanoseconds per pixel, using an In-lens SE detector, and can acquire very large volumes within a reasonable time. In contrast, images from tissue sections processed with the TOLA protocol require more time to acquire, but these samples have well-preserved ultrastructure suitable for the analysis of synapses.

The MultiSEM has a total current of 35 nA (61 beams at 570 pA per beam). While with single beam set-ups we use 200 pA to 3.2 nA, the MultiSEM uses 10 to 175 times more current. In addition, the stage bias of 28.5 keV is also much higher than in a single beam microscope. Both beam current and stage bias require excellent conductive properties of the tape. Practically, we observed cases where commercially available cc-Kapton tape works well in single beam SEM but became charged in MultiSEM (cc-Kapton tape with surface resistance of 6530 MΩ □$^{-1}$). From this point of view, we believe that the CNT tape, which has a surface resistance of about 240 Ω □$^{-1}$, provides improved

imaging conditions for MultiSEM, which are not always guaranteed with the cc-Kapton tape.

The conductive nature of CNT tape is advantageous for imaging with high doses, such as imaging with longer dwell times, with large probe currents, or with repeated imaging. Imaging with longer dwell times may give better quality images. The larger probe current, which can be adjusted without significant beam size changes in the latest SEM, may allow for shorter dwell times, resulting in high-throughput imaging. Repeated imaging allows recapturing of the same area for further imaging and tomography analysis using various acceleration voltages.

We believe that our CNT tape and its related staining protocol will facilitate life science research on tissue ultrastructure. For instance, the high-resolution images allow easy and reliable identification of both asymmetrical and symmetrical synapses and will therefore promote connectomics research[32]. The method also offers reliable measurement and analysis of nanoscale cellular structures, such as synaptic junction areas and related cellular structures[6,51], which could provide morphological evidence for dynamic changes in neural microcircuitry during learning and plasticity[52–56]. The visualization of individual myelin sheaths or mitochondria may also promote a deeper understanding of related pathological or etiological sources of brain diseases[57–59].

A distinctive advantage of the ATUM-SEM method is its feasibility for large image acquisition, further compounded by stitching together many image tiles. The current largest single image size using Sigma/Atlas5 (Fibics incorporated, Ottawa, Canada) is $32,000 \times 32,000$ (1024 MB) pixels. The area for this image size can be $160 \times 160\,\mu m$ with 5 nm pixel$^{-1}$, appropriate for contouring synapses, or $800 \times 800\,\mu m$ with 25 nm pixel$^{-1}$, appropriate for dendrite and axon tracing. The former pixel size allows us to identify synaptic contacts and small neural structures among hundreds of synapses found in the imaged area, whereas the latter allows the identification of dendritic and axonal arborization patterns as large as the whole dendritic field of a layer 5 pyramidal cell in the rat cortex. After the identification of a region of interest on a dendritic tree in a low magnification image, we can subsequently image it in high magnification for detailed synapse observation using ATUM-SEM. In this way, microcircuit architecture can be systematically studied. A whole mouse brain can be imaged serially with 10 nm pixel$^{-1}$ using stitched image tiles with ATUM-SEM using CNT tape, as shown in Fig. 7 [22,60].

The next technical challenge in the ATUM-SEM method is how to process the serial image dataset of such a large size for 3D reconstruction analysis. This question is composed of individual steps in tissue processing, imaging, aligning the serial images, segmenting an area of interest, and analyzing the neural elements found therein. We found the 1024 MB ($32,000 \times 32,000$ pixels) size images can be aligned using Fiji registration plugins[61] (Register Virtual Stack Slices) or TrakEM2, but we still await new tools for automated segmentation[5,16,62] for 3D reconstruction, a current workflow bottleneck.

In conclusion, the CNT tape and staining protocol reported here will improve ATUM-based serial section EM data collection and facilitate brain microcircuit analysis.

## Methods

**Ethics statement**. All surgical and animal care methods were performed in strict accordance with the Guidelines for the Use of Animals of IBRO and our institutional Animal Care and Use committee of the National Institute for Physiological Sciences. Every effort was made to minimize suffering.

**Tissue preparation**. Five male Wistar rats (6–8 weeks of age, 140–160 g) were anesthetized with an overdose of pentobarbital (60–75 mg kg$^{-1}$) and perfused through the heart with 5–10 ml of a solution of 250 mM sucrose, 5 mM MgCl$_2$ in 0.02 M phosphate buffer (pH 7.4) (PB), followed by 300 ml of 4% paraformaldehyde

containing 0.2% picric acid and 1–2% glutaraldehyde in 0.1 M PB[63]. Brains were then removed and oblique horizontal sections (50 μm thick) of frontal cortex and striatum were cut on a vibrating microtome (VT1200S, Leica Microsystems, Wetzlar, Germany) along the line of the rhinal fissure. Tissue sections were put in the cryoprotectant solution: 30% glycerol, 30% ethylene glycol, 0.04 M PBS[64] in a glass small screw vial (5–10 ml volume size) and stored in a freezer at −20 °C or −30 °C until use.

**Metal post-fixation and preparation for EM observations**. For twice osmium protocol (TO), the sections were washed in 0.1 M PB and post-fixed for 1 h in 1.5% potassium ferrocyanide and 1% osmium tetroxide in 0.1 M PB, followed by 1 h in 1% osmium tetroxide in 0.1 M PB[63]. The sections were washed 3 times for 10 mins each with 0.1 M PB between the staining steps. After washing in Milli-Q water, they were dehydrated in graded dilutions of ethanol with 1% uranyl acetate added at the 70% ethanol dehydration state.

For twice osmium plus lead aspartate protocol (TOLA), the sections were washed in 0.1 M PB and post-fixed for 1 h in 1.5% potassium ferrocyanide and 1–2% osmium tetroxide in 0.1 M PB. Sections were washed with 0.1 M PB followed by 1 h in 1–2% osmium tetroxide in 0.1 M PB. Then the sections were washed in Milli-Q water and placed in 1% uranyl acetate (aqueous) at 4 °C protected from light overnight. On the next day, Walton's en bloc lead aspartate staining was performed[65]. The lead aspartate solution was prepared by dissolving 0.66 g of lead nitrate in 10 ml of 0.03 M aspartic acid, pH adjusted to 5.5 with 1 N potassium hydroxide and kept in an oven at 60 °C for 30 min until dissolved. Sections were washed with Milli-Q water several times and placed in the solution in the oven for 30 min. The sections were again washed with Milli-Q water and dehydrated in graded dilutions of ethanol.

For modified heavy metal staining (mHMS) histology protocol, the sections were washed in 0.1 M PB and post-fixed for 1 h in 1.5% potassium ferrocyanide and 2% osmium tetroxide in 0.1 M PB, followed by fresh 1% thiocarbohydrazide (TCH) solution for 20 min to bridge the following osmium tetroxide impregnation[38]. The sections were incubated for 1 h in 2% osmium tetroxide solution, then placed in 1% uranyl acetate (aqueous) at 4 °C protected from light overnight[37,41,42]. On the next day, Walton's en bloc lead aspartate staining was performed as described above and dehydrated in graded dilutions of ethanol. The sections were washed a few times with Milli-Q water between the steps.

Sections were flat-embedded on silicon-coated glass slides in epoxy resin, Durcupan ACM (Sigma-Aldrich, St. Louis, USA) or EMbed-812 (#14120, Electron Microscopy Sciences, Hatfield, PA, USA) and placed in a 60 °C oven for 48–72 h for polymerization. Following the re-embedding of tissue samples, they were serially re-sectioned into 40–50 nm-thick ultrathin sections, and they were collected using an ATUMtome (Boeckeler Instruments, Inc., Tucson, USA) or a conventional ultramicrotome (Reichert Ultracut S, Leica Microsystems, Wetzlar, Germany). The tissue sections with the rOTO treatment or twice osmium plus lead aspartate treatment were observed by the SEM without any further section staining, but some of the tissue sections with twice osmium treatment were also stained with lead citrate or a 1% uranyl acetate solution followed by lead citrate for better contrast. The lead citrate solution[66] was prepared by dissolving 1.33 g of lead nitrate and 1.76 g of sodium citrate in 30 ml of Milli-Q water and stirred for 30 min. We then added 8 ml of sodium hydroxide and diluted to 50 ml with Milli-Q water. The cloudy solution cleared.

**SEM observation**. Serial ultrathin sections on tape were cut into strips and mounted in order on 4-inch silicon wafers (number of dust particles < 100, resistance 1–30 Ω, Canosis Co. Ltd., Tokyo, Japan) with double-sided adhesive conductive tape (carbon conductive double-faced adhesive tape with a nonwoven fabric core, Nisshin EM Co., Ltd., Tokyo, Japan). To ground the conductive layer on the tape surface to the wafer, we put copper foil tape (Takeuchi Kinzokuhakufun Kogyo, C1020R-H-40um, Supplementary Figure 2) on the edge of the tape and wafer. The sections were observed using In-lens SE or BSD with SEM (Sigma, Carl-Zeiss Microscopy GmbH, Oberkochen, Germany). We also used a SEM equipped stage bias potential (Gemini300/500, Carl-Zeiss Microscopy GmbH, Oberkochen, Germany) using the In-lens SE or the BSD optimized for low acceleration voltage (OnPoint BSD, Gatan, Inc., Pleasanton, CA, USA). We also used Atlas 5 (Fibics incorporated, Ottawa, Canada) for large area imaging.

**Mouse whole-brain SEM observation**. The mouse whole-brain was histologically prepared using a BROPA-like protocol (brain-wide reduced osmium staining with pyrogallol-mediated amplification)[22,67]. More than 1000 serial sections were cut using a 6 mm ultra 45° knife (DiATOME, Biel, Switzerland) with a nominal cutting thickness of 100 nm and 1.0 mm s$^{-1}$ cutting speed and collected onto plasma-treated CNT tape using the ATUMtome. The tape was then cut into strips and attached onto a 4-inch silicon wafer (SC4CZp-525, Science Services GmbH, Munich, Germany) using 25 mm wide double-sided adhesive carbon tape (P77819-25, Science Services). Imaging used a custom silicon-diode BSD (AXUV, International Radiation Detectors) equipped in a SEM (QuantaFEG 200, FEI, Hillsboro, Oregon, USA).

**MultiSEM observation**. The sample (cortical mouse brain tissue, primary soma-tosensory cortex) was histologically prepared for EM with a modified rOTO staining protocol designed for large-volume en bloc staining [41] (Fig. 7f, g and Supplementary Figs. 6, 11, 12) and mHMS protocol (Supplementary Fig. 6). Serial ultrathin sections were collected on plasma-treated CNT tape using the ATUM-tome. The slices were cut using a 3 mm ultra35° knife (DiATOME, Biel, Switzerland) with a nominal cutting thickness of 35 and 25 nm respectively, and 0.3 mm s$^{-1}$ cutting speed. The tape was glued onto a 4-inch silicon wafer (SC4CZp-525, Science Services GmbH, Munich, Germany) using 25 mm wide double-sided carbon tape (P77819-25, Science Services). The edges of the CNT tape were additionally connected using 8 mm wide double-sided carbon tape (P77816, Science Services GmbH). Secondary electron imaging was performed on a Zeiss MultiSEM 505 (61 beams).

**TEM observation**. Serial ultrathin sections were serially sectioned at thickness 50 nm with an ultramicrotome (Reichert Ultracut S, Leica Microsystems, Wetzlar, Germany). Ultrathin sections were mounted on Formvar-coated single-slot grids. EM images of labeled axon terminals and dendrites were captured with a CCD camera (XR-41, Advanced Microscopy Techniques, USA) in Hitachi H-7000, and HT-7700 EMs (Hitachi High-Technologies, Tokyo, Japan).

**Tape**. We checked many kinds of tapes, copper foil (20 μm or 40 μm thickness): Takeuchi Metal Foil & Powder Co., Ltd.; 8 mm video tape (11 μm or less thickness): Sony, Tokyo, Japan; open-reel tape (53 μm thickness): Studio Master 911, Recordable Media Group International B.V., Oosterhout, The Netherlands; ITO (indium tin oxide) coated PET (polyethylene terephthalate) (50 μm thickness): FLECLEAR, TDK Electronics Co., Ltd., Tokyo, Japan, ELECLEAR, Teijin limited, Tokyo, Japan; CNT (double-walled carbon nanotube)-coated PET tapes (50 μm thickness): carbon nanotube transparent conductive film, Toray Industries, Inc., Tokyo, Japan; Kapton tape (polyimide film), DuPont, Wilmington, USA, the carbon coating on the Kapton tape was applied about 10 nm thick using a JEE-400 Vacuum evaporator, JEOL Ltd., Tokyo, Japan; cc-Kapton tape (polyimide), Boeckeler Instruments, Inc., Tucson, USA.

**Plasma treatment**. The CNT-coated PET tape surface was originally hydrophobic. For hydrophilization, the tape surface was treated with plasma glow discharge using a unique slit type atmospheric pressure plasma generator (A-1000, SAKIGAKE-Semiconductor Co., Ltd., Kyoto, Japan). A roll of tape is usually supplied with a three inch core. We set the roll of tape on a reel for a three-inch core and an empty ATUM one inch core reel was set on a reel drive for rewinding (Supplementary Fig. 1). The tape end was put on the empty one inch core center and rewound from the three-inch core reel to the one-inch core reel. Then the rewound tape roll was moved to the one-inch core reel and the plasma slit torch (Supplementary Fig. 1) was set between the reel drive and the one-inch core reel (Supplementary Fig. 1). The tape reeled out from the one-inch core reel and wound to the empty one-inch core reel set on the reel drive, passing underneath the plasma slit torch (Supplementary Fig. 1). The plasma was irradiated on the running (10 mm s$^{-1}$) tape surface using a custom reel-to-reel motorized winder (Supplementary Fig. 13) under expelled nitrogen gas (0.6 MPa). The PWCA of the CNT-coated PET tape was originally 79.5 degrees and it became 7.4 degrees after the plasma coating (Fig. 1k). The PWCA was measured with a contact angle meter (DMs-200 or CA-X, Kyowa Interface Science Co., LTD., Niiza, Japan). We tested the plasma treatment effect using ultrathin sections and found that the sections were collected without any wrinkles on the treated tape and the effect of the plasma treatment lasted at least 13 months.

**GABA post-embedding immunohistochemistry**. Ultrathin sections of 70–90 nm thickness of rat cortex were collected on either CNT or cc-Kapton tape with the ATUMtome. We applied post-embedding GABA immunohistochemistry to the serial ultrathin sections of the brain tissue processed with TO and TOLA on the CNT tape and cc-Kapton tape. The ultrathin sections were washed with 0.05 M TBS containing 0.1% Triton-X (TX) and incubated with rabbit antiserum against GABA (1:2500 or 1:5000; A-2052, Sigma-Aldrich, St. Louis, USA) in TBS containing 0.1% TX overnight. The ultrathin sections were then incubated with 15 nm colloidal gold conjugated anti-rabbit IgG (1:200; BBInternational #GAR15, Cardiff, UK) overnight in TBS containing 0.1% TX, and stained with 1% aqueous uranyl acetate followed by lead citrate. Images of labeled axon boutons and dendrites in the sections on the tape were captured with the SEM. Quantitatively, synaptic boutons could be divided into two classes on the basis of gold particle densities. Particle densities were greatly different in GABA-negative and GABA-positive terminals: $2.0 \pm 3.8$ μm$^{-2}$ ($n = 245$) and $59.7 \pm 18.9$ μm$^{-2}$ ($n = 48$), and GABAergic terminals were defined as terminals with a gold particle density above 30 particles μm$^{-2}$. In serial ultrathin sections, presynaptic GABA-negative or GABA-positive boutons always showed similar colloidal gold density in multiple sections. We also applied Alexa 594-conjugated anti-rabbit IgG secondary antiserum (1:200; A11012, Thermo Fisher Scientific, Waltham, USA) followed by the primary anti-GABA antiserum on 200 nm thick serial sections on the CNT tape and images were captured with a fluorescence microscope (Olympus BX60, Tokyo, Japan).

**Electron dose**. The electron dose was defined by the equation below[31].

electron dose (e$^{-}$ nm$^{-2}$) = beam current (amperes) × (1/1.60217657 × 10$^{-19}$ (coulombs /electron)) × pixel dwell time (seconds)/pixel size$^2$ (nm)

**Surface resistance measurement**. Surface resistance was measured with a resistivity meter (Loresta-AX MCP-T370, measurement range 10$^{-2}$–10$^6$ Ω □$^{-1}$, Mitsubishi Chemical Analytech Co. Ltd., Chigasaki, Japan). The resistance of the cc-Kapton tape with high resistance (>10$^6$ Ω □$^{-1}$) were measured with a semiconductor characterization system (Keithley 4200-SCS, Tektronix, Beaverton, OR, USA) and the values were converted to sheet resistances.

**Tape specifications**. The emission spectrum was analyzed with a laser confocal microscope (LSM 880, Carl Zeiss Microscopy GmbH, Oberkochen, Germany) using a Plan-Apochromat 20× lens, 2.0% laser power for excitation wavelength 405 nm with a 30 mW diode, 5.5% laser power for excitation wavelength 488 nm with 25 mW argon, 2.0% laser power for excitation wavelength 561 nm with 20 mW DPSS, and 20% laser power for excitation wavelength 633 nm with 5 mW HeNe.

Absorption and transmittance spectra were measured with a spectrophotometer (U-4000, Hitachi High-Technologies, Tokyo, Japan).

Haze and total light transmittance were measured with a haze meter (NDH 200 Haze Meter, Denshoku Industries Co. Ltd, Tokyo, Japan)

**Signal efficiency for a 50 nm thick section**. Proportion of BSEs reflected only from the 50 nm-thick section among all BSEs generated from within the section.:

$$\text{SE-50 nm}(\%) = (BSE0 - BSE50) \times 100/BSE0$$

where

SE-50nm is signal efficiency for a 50 nm thick section
BSE0 is number of BSEs in 0–10 nm bin
BSE50 is number of BSEs in 50–60 nm bin

**Estimation of the conductivity**. We calculated conductivity using this formula.:

$$\sigma = 1/(R_s \times t)$$

where

$\sigma$ is conductivity with SI units of siemens per meter (S m$^{-1}$)
$R_s$ is sheet resistance with SI units of ohms per square (Ω □$^{-1}$)
$t$ is film thickness with SI units of meter (m)

**Contrast**. Contrast was calculated using the Michelson contrast formula with values at half width of the intensity histogram shown in Fig. 6.

Contrast $= (L_{max} - L_{min})/(L_{max} + L_{min})$ where $L_{max}$ is maximum gray scale value at half width of the histogram
$L_{min}$ is minimum gray scale value at half width of the histogram
An image with good contrast has values close to 1.
An image with low contrast has values close to 0.

**Contrast-to-noise ratio (CNR)**. We hypothesized the intensity histogram of electron micrographs (Fig. 6) was composed of two independent constituents; membrane (black in an inverted positive image) and background (white in an inverted positive image). The number of detected electrons with the SEM detector was also composed of the two constituents. Intensity values of their peaks are Nm and Nb, and the variance of noise are σm$^2$ and σb$^2$, respectively.

We calculated contrast value ($C$):

$$C = Nb - Nm$$

We adjusted contrast and brightness during imaging, so Nm and Nb were modified with amplification ($a$) and offset ($o$). Intensity values of the peaks in the image intensity histogram: Vm and Vb, and the variance of noise σvm$^2$ and σvb$^2$, were calculated:

$$Vm = aNm - o, \ Vb = aNb - o, \ \sigma vm^2 = a\sigma m^2, \ \sigma vb^2 = a\sigma b^2$$

The ($C$) was converted using those formulas:

$$C = (Vb - Vm)/a$$

For a calculation of contrast-to-noise (CNR)[44,45], the background noise value ($\sigma vb^2$) is used as a noise value, then the CNR can be obtained:

$$CNR = \frac{(Vb - Vm)/a}{\sigma vb^2/a}, \text{ then } CNR = \frac{(Vb - Vm)}{\sigma vb^2}$$

For a calculation of CNR of the intensity histograms obtained with the BSE and In-Lens SE, we performed a double-Gaussian fit to the histograms to extract the intensity values of the peaks (Vm, Vb) and the variance of individual Gaussian distributions ($\sigma vm^2$, $\sigma vb^2$) using the following equation:

$$f(x) = A \exp\left(-\frac{(x - Vm)^2}{2\sigma vm^2}\right) + B \exp\left(-\frac{(x - Vb)^2}{2\sigma vb^2}\right)$$

where $A$ and $B$ are number of pixels in the peaks.

The CNR is believed to be absolute and can therefore be used for comparisons between different values obtained under varied acceleration voltages[44,45].

Electron micrographs and their raw data for the signal intensity histograms in Fig. 6 are available[70].

**Data availability**. The data that support the findings of this study are available from the corresponding authors upon reasonable request.

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

## Acknowledgements

We thank Mrs. Sarah Mikula, Dr. Charles Yokoyama, Dr. Kazue Semba and Dr. Moritz Helmstaedter for valuable comments, Mr. Pat Brey, Boeckeler Instruments, Inc., Dr. Robert Kirmse, Dr. Akio Sekigawa, Carl Zeiss Microscopy, Mr. Shigeaki Tachibana, Dr. Mami Konomi, Hitachi High Technologies, for valuable comments. Dr. Anna Lena Eberle, Dr. Tomasz Garbowski, Carl Zeiss Microscopy, for providing the MultiSEM image in Supplmentary Fig. 6e and valuable comments. Dr. Liu Ning and Dr. Huang Jianliu, Carl Zeiss Microscopy, for providing the SEM images in Supplementary Figs. 1c, 7e. Dr. Mitsuo Suga, JEOL Ltd., for valuable comments on how to obtain the CNR values in Fig. 6h. Mr. Takashi Oi, Toray Industries, Inc. and Mr. Osamu Watanabe, Toray Advanced Film, Co., Ltd., for the CNT film supply and surface resistance measurement. Mr. Yuzuru Adachi, Sakigake semiconductor, Co., Ltd., for the analysis of PWCA of the tape. Mr. Haruhiko Itoh, Teijin Limited, for the analysis of PWCA and transmittance of the tape, Mr. Akira Sato, Carl Zeiss Microscopy, for analysis of the emission spectrum of the tape, Mr. Koji Yamao, Keyence corp. for surface profile measurements of the SEM beam scanned tape. This work was supported by JSPS KAKENHI (No. 25250005, 25290012, 15K14324, 17J04137), and MEXT KAKENHI on Innovative Areas "Adaptive circuit shift (No. 3603)" (No. 26112006, 15H01456), and on "Brain information dynamics underlying multi-area interconnectivity and parallel processing (No. 4905)" (No. 17H06311). The NOVARTIS Foundation (Japan) for the Promotion of Science, The Okazaki ORION project, and The Imaging Science Project of the Center for Novel Science Initiatives (CNSI), National Institutes of Natural Sciences (NINS) (No. IS291001).

## Author contributions

Y.Ku. conceived the study, designed the experiments, analyzed and interpreted the data, generated the figures; Y.Ku., S.H. performed the tissue preparation, acquired the data and micrographs; J.So. performed the simulation; M.S., J.St.., A.G. acquired the micrographs for Fig. 7f, g and Supplementary Figs. 6f, 11, 12; R.N. acquired the micrographs for Fig. 5e, f, k, l, Supplementary Figs. 6c, d, 10; T.M. acquired the CNR values; S.M. acquired the micrographs for Fig. 7a-e; Y.Ku., J.So., M.S., J.St., S.M., Y.Ka. drafted and revised the manuscript.

## Additional information

**Competing interests:** The authors declare no competing financial interests.

