## [Peer Review File · Nature Communications]

Reviewers' comments:

Reviewer #1 (Remarks to the Author):

Kubota et al

A novel tape material and staining protocol for volume electron microscopy applications

Summary:

This manuscript by Kubota and colleagues describes a new tape for ATUM/volume electron microscopy with advantages compared to the currently standard Kapton tape. The work describes a range of tests done on potential substitute tapes (mostly selected to have higher conductivity than Kapton in order to avoid additional coating steps), and then a more detailed characterization of the CNT-PET tape that emerges as the best candidate as a substitute. The authors describe a procedure (including a custom-built device) for surface-treatment to reduce hydrophobicity of the tape and a comparison of staining/imaging conditions to optimize use of the tape for imaging neuropil.

Evaluation

This manuscript describes a valuable technological addition to the emerging ATUM technology. There are, however, a number of issues (see below) that should be addressed to improve the manuscript, with the most significant being that (1) more direct quantitative comparison between Kapton and CNT-PET would buttress certain claims; (2) the advantage of compatibility with light microscopy for CNT-PET should be supported by data; (3) all aspects of the new procedure should be documented in detail (e.g. construction of the plasma glow discharge device; the final, optimized staining and imaging protocol) and (4) that the manuscript (and the illustrations) could benefit from being restructured and revised to flow more naturally. Given such revisions, the paper in my view would be an important and impactful contribution to the fast growing body of literature on semi-automated volume electron microscopy.

Major points:

1) The selection of the different tapes is not explained very well – it remains unclear, to what degree their characteristics (e.g. variable thickness) are determined simply by availability; also, whether and how easily they would be commercially available is an important piece of information. A table that sums up all this information – reason for choice, availability, results with a given tape as a supplementary could really help the reader. If possible, some information about cost would also be interesting.

2) Line 165 “we used a unique slit type atmospheric pressure plasma generator (A-1000, SAKIGAKE-Semiconductor Co., Ltd., Kyoto, Japan) and a custom-made reel-to-reel motorized winder (Fig. 1i, j).” – the need for this device (and potentially its custom-design) will be an important factor in the degree to which this method is adapted, in addition to the availability of the tape. The authors need to make CAD-design plans for the winder and electrical control plans for the plasma generator available, so that other labs can readily adopt the technique.

3) Line 261: “Applicability in LM: The CNT tape has no auto-fluorescence in red or infrared light, and only very faint auto-fluorescence in green and blue. Total light transmittance is more than 90% (Supplementary Fig. 1f), therefore the tape may be used for multi-color fluorescent immunohistochemistry, as used in array tomography.” This is an important point and potentially a big advantage of this tape. So the authors need to provide substantial data to characterize the light microscopic properties of this tape such as: Transmission/absorption spectrum; emission spectrum; refractive index etc. – all compared to coated Kapton. Ideally a proof-of-principle experiment using light microscopy should be shown.

4) The segment about the modified staining protocol comes a bit out of the blue and does not seem absolutely essential – as in contrast to the rest of the paper, it refers more to a specific application (identification of synapses). Still, this will address the interest of a significant number of potential readers of this work, but if kept it needs a bit of a segue and some revision (lines 314-328 are too much information on already published results, so they can be shortened); also, a more systematic presentation should be adopted (e.g. in Fig 5, the TO only has to be shown to appreciate the effect of the extra heavy metals). Overall, as is, filled with similar-sounding acronyms and a somewhat unorganized back-and-forth (TO is introduced as a comparison, before it is explained; TOLA suddenly shows up combined with lead nitrate etc.), this segment is not really easy to follow. Again, a schematic of tabulated comparison of the protocols used would help and should be accompanied by a more systematical explanation of the three staining protocols (SBF, TO, TOLA) – and how TOLA derives from SBF and TO. Moreover, there is no explanation given on the fact that TOLA applies lead aspartate en bloc instead of lead nitrate in sections (which, however, can be added, see Fig 5 e,j).

Minor points:

1) Line 151: "The double-walled CNT tape shows superior surface conductivity and transparency over single-walled or multi-walled CNT versions." – please include data to buttress this point. Overall the paper would benefit from adding quantifications wherever possible, comparing at least the Kapton and the CNT-coated tape, such as a quantification of the major defect frequency – scratches, wrinkles, etc. Similarly, the statements about tape resilience after chemical and physical treatments could be supported by images and data derived from imaging empty films.

2) Line 176: "The plasma-treated CNT-PET tape hydrophilization lasts up to 7 months and it does not affect the surface conductivity, whereas plasma-treated Kapton tape hydrophilization lasts only a few weeks and reduces the surface conductivity because of clean-up the coated carbon particles." This is important information and speaks to the possibility of providing ready-to-go tapes for potential users – so the data should be shown.

3) Line 181: "(87.3 % reduction of conductivity; 257 ± 4.5 ohms/square without plasma discharge treatment vs 284 ± 18.1 ohms/square with plasma discharge treatment, $n=3$ each)." At this point I got a bit confused: Which numbers were used to calculate the "87.3%"? It is a bit unclear what was compared here and how this relates to the numbers given in Line 147ff. Also, what exactly is the unit here (I do not recognize "square" as a unit)?

4) Line 185ff.: The segment on Monte Carlo Simulations of track depth seems displaced, as it refers to the open reel, which the section has already bypassed. If at all necessary (the images Fig 2g-q, without a quantitative analysis – i.e. number of tracks that crosses 50nm etc. – and comparable scaling of the axes really do not help the reader much), these results should be incorporated in the tape screen section Line 111ff. Alternatively it can be shifted back – Line 214ff – to discuss the depression and its possible dependence on primary electron penetrating into the polymer (Note: This notion could be tested by imaging thicker sections and seeing whether the threshold for the formation of a depression shifts to the higher EHT values predicted by the simulations).

5) Line 214ff: "(Supplementary Fig. 5-8)" – as these belong together, could the results be organized into a single figure – perhaps even including SFig 4? The current design unnecessarily inflates the length of the supplement.

6) SFig 11: After how many exposure reiterations does the image quality deteriorate?

7) Line 362ff: This seems like a 'loop' in the flow of the paper, back to previous topics and figures –

perhaps this information can be consolidated in one place?

8) Staining paragraph, line 297ff: The Hua et al., Nat Comm, 2015 paper (also cited in this article) introduced a staining protocol that separates OsO₄ penetration from ferrocyanide enhancement. This avoids unsaturated lipid depletion but allows membrane penetration. Additionally, Hua et al. established a two-step temperature protocol for UA impregnation that exploits the fact that diffusability and reactivity of UA is differentially affected. This method shows very good contrast and penetration. The authors are not stating why they did not consider these steps.

9) There is quite a bit of grammar to fix – this manuscript needs a thorough revision in this regard.

10) The figure sequence and the panel order in the figures would be reorganized to a more consistent stream to avoid back-and-forth for the reader.

11) Overall, a broad range of sample, staining and detector types were used in this study. It was not obvious to me for all the figures what was the sample and the staining used. I would suggest to also organize this in a table or in a systematic fashion in the legends, so the reader can ensure s/he is only comparing what can be compared.

Reviewer #2 (Remarks to the Author):

The paper by Kubota et al presents a new supporting tape to be used for collecting ultrathin sections cut with the automated tape collecting ultramicrotome (ATUM). This microtome system has significant potential for the field of connectomics where very large volumes of brain tissue are needed to be imaged for the mapping of neural circuits. The automated collection of sections onto a tape, that is then imaged in the scanning electron microscope, reduces the risk of interruptions to the serial sections.

The authors show that the use of Kapton tape as suggested in the original work by Hayworth et al. is not ideal for two reasons. Firstly there is significant folding of the sections, and secondly, there is insufficient charge removal, and therefore too many imaging artefacts. This part of the paper is slightly confusing as although the authors in the current work show images of folds on sections, this was certainly never brought up in the original paper. The original paper only mentions that folding is rare. It is also noted that the original paper uses a conductive coat on the Kapton tape to reduce the charging, so I am not sure why the current paper even uses the uncoated Kapton tape as a comparison.

My other comment relates to the second aspect of the paper, which is the sample preparation and imaging conditions for this type of microscopy. They are as follows:

In figure 2, the authors look at the results obtained with magnetic tape as a support and try to explain the poor imaging results with the use of monte carlo simulation. They explain this in terms of how the electrons in the higher energy beam are interacting deeper in the section and probably with the magnetic surface. However, I am troubled as to how they can interpret this when the imaging is done with the secondary electrons, that emerge very close to the surface, and are unlikely to traverse through significant thicknesses of resin. If electrons of any type are interacting with magnetic surface then there will be other effects. I would suggest that a magnetic surface will effect how the electrons interact with the section, and a simple Monte Carlo simulation, such as the one shown, is not sufficient to explain this, and I do not agree with the interpretation. How is it that primary electrons, so deep in the section, will cause the production of secondary electrons giving information about the magnetic

surface?

In Figure 3 the authors show the effect of the electron beam on the section. It is not clear why this is relevant to the paper, or how the authors are able to interpret this as a depression in the section. How was the depth of this depression measured? If it is a depression, as the authors contend, then how can they be sure this is through mass loss, and not some heating effect of the beam.

Another general aspect of the paper that concerns me is that it is clearly aimed at providing the best quality of images. The authors give a description as to why they used the conditions for clearly seeing the synaptic connections. However, I notice that the images are collected with pixel dwell times that are significantly longer than were used in the original paper. Hayworth et al used very short imaging time because this is critical for being able to image large volumes in a reasonable period. It may be that synaptic connections are not imaged in optimal conditions, but the Hayworth paper explains that across serial sections the identification is straightforward. Therefore, I am concerned that the current piece of work is ignoring the fact that using their technique imaging times for a significant volume would be excessive, or even completely unfeasible.

There are some minor comments as well. There are instances where the authors use acronyms that are not necessary, or confusing. For example the acronym SBF is used for the staining protocol, but this is confusing when many understand it as 'serial block face', for describing an imaging technique. The acronym EMg for electron micrograph is also strange as in many cases they could use the word, 'image'.

The paper describes the use of a number of different tapes for holding the sections, but are briefly described as not being suitable. It's not clear why these are explained in the paper if they give no advantage. It's also not clear why it's necessary to provide an image of them.

In summary, this paper gives some useful information as to the preparation and imaging of brain tissue collected onto tape for imaging with SEM. However, there are so many different variables given it's hard for the reader to understand the logic, and which are the most important. The paper presents different support films, conductive layers, stains, microscopes, voltages, dwell times, imaging modes, but does not give thorough comparisons. The paper needs significant reorganisation to convey the most important details.

Reviewer #3 (Remarks to the Author):

The paper addresses a current need for the growing number of fields that require three dimensional electron microscopy. The ATUM tape-to-SEM technology requires a stable substrate that satisfies many chemical, mechanical, and environmental criteria. To date most tape materials satisfy a subset of requirements, leaving a researcher with suboptimal performance and research results. After performing an extensive search and testing using a variety of electron microscope techniques, the authors found a carbon nanotube coated substrate that meets all the requirements of a successful tape material. The demonstration of the GABA immunolabeling helps complete the entire repertoire of how the tape may be used. The explanation of the CASINO code is a weak point in the paper and should be revised.

Line 101 Should be a 24% reduction in conductivity $\{(3367-4172)/3367\}$

Line 304 Striped pattern not obvious in supplemental 3c,d

Line 185 through line 200: This paragraph is poorly executed. The concepts of interaction volume, range, straggle and escaped depth should be discussed in the context of the section thickness and underlying substrate. If this is not done, the paragraph should be removed and figure 2 reworked.

Lines 210 and 211: Can the effects of mass loss be expounded upon? How will the depressions affect downstream efforts such as the stitching of mosaics and the 3D alignment of the serial sections.

Line 223: Linking the cause of the depression to additional crosslinking of the overcoat layer is made; however, earlier, the process of mass loss was used to describe the reason for the depression. Please better explain what is causing the depression. Given the importance of the CNT/overcoat layer, the overcoat layer polymer should be identified along with the thickness of the layer.

There does not seem to be a distinction made between using a SE2 detector and an in-lens detector (Explain why figure 3c does not look like figure 2 u)

Figure 1: The total tape thickness is not defined. Figure 1i does emphasize the control panel, but not the reel-to-reel or plasma system.

Figure 3 SMST is not defined.

Figure 4 The images in this figure 4 d and e, should be denoted as single-beam backscattered images, as distinct from the mSEM secondary electron images

Figure 5: The molecular formula for lead nitrate(II) is $Pb(NO_3)_2$. Using 'In' is not very informative. If the abbreviation is made for lack of space, maybe use PbN.

Supplementary Figure 3 does not contain an example image of carbon-coated Kapton tape.

Supplementary Figure 4: Can you comment on the physical mechanism of the depression?

Supplementary Figure 5: I am not sure that I understand the point of this figure. To begin we are not told that the working distance was optimized to collect an image with the largest signal to noise ratio. Second, the images shown are more a function of the backscattered electron detector properties, rather than a collective property of the other imaging parameters such as, EHT or dwell time. For instance the energy response function of the BS detector will determine the SNR for a given EHT, while the bandwidth of the detector will determine the amount of noise in the image for a given dwell time. So saying that 4 and 5 kV are acceptable while 3 and 6 kV are unacceptable is only accurate for this specific backscatter detector and not a general result.

Supplementary Figure 6: The dose rate (#electrons/unit area/time) and the total electron dose (#electrons per unit area) for each condition should be stated. Though the image suggests that increasing the EHT increases the depression depth, the evidence of the depression depth not changing with dwell time is lacking. If the authors want to make this claim, the depression depths should be measured.

Supplementary Figure 7: The dose rate (#electrons/unit area/time) and the total electron dose (#electrons per unit area) for each condition should be stated. The type of SE detector should be described (Everhart-Thornley or in-lens). It is unlikely that the darker center regions result from surface depressions. More likely is that these are focusing squares and the darkening is the result of a thin layer of carbon (adventitious) build up. In addition, a contrast reversal is apparent in the 1kV images is seen. This should be acknowledged and commented upon. Again there is no comparison to

the carbon-coated Kapton.

Supplementary Figure 8: The SE images now clearly show that the scan depressions not only are a function of accelerating voltage, but also a function of dwell time. This observation is made possible by the surface sensitivity of secondary electrons. This means that the conclusion in Supplementary Figure 6 is probably incorrect.

Supplementary Figure 9: Imaging parameters for all images should be described. In general the images in this figure have poor signal to noise ratio and narrow histograms. Higher quality images at higher magnifications (structures of interest should be easily seen) should be acquired.

Reviewers' comments and our replies:

Reviewer #1 (Remarks to the Author):

Kubota et al

A novel tape material and staining protocol for volume electron microscopy applications

Summary:

This manuscript by Kubota and colleagues describes a new tape for ATUM/volume electron microscopy with advantages compared to the currently standard Kapton tape. The work describes a range of tests done on potential substitute tapes (mostly selected to have higher conductivity than Kapton in order to avoid additional coating steps), and then a more detailed characterization of the CNT-PET tape that emerges as the best candidate as a substitute. The authors describe a procedure (including a custom-built device) for surface-treatment to reduce hydrophobicity of the tape and a comparison of staining/imaging conditions to optimize use of the tape for imaging neuropil.

Evaluation

This manuscript describes a valuable technological addition to the emerging ATUM technology. There are, however, a number of issues (see below) that should be addressed to improve the manuscript, with the most significant being that (1) more direct quantitative comparison between Kapton and CNT-PET would buttress certain claims; (2) the advantage of compatibility with light microscopy for CNT-PET should be supported by data; (3) all aspects of the new procedure should be documented in detail (e.g. construction of the plasma glow discharge device; the final, optimized staining and imaging protocol) and (4) that the manuscript (and the illustrations) could benefit from being restructured and revised to flow more naturally. Given such revisions, the paper in my view would be an important and impactful contribution to the fast growing body of literature on semi-automated volume electron microscopy.

Major points:

1) The selection of the different tapes is not explained very well – it remains unclear, to what degree their characteristics (e.g. variable thickness) are determined simply by availability; also, whether and how easily they would be commercially available is an important piece of information. A table that sums up all this information – reason for choice, availability, results with a given tape as a supplementary could really help the reader. If possible, some information about cost would also be interesting.

We described requirements for the ATUM tape in the last paragraph (page 4 line 98~) for better understanding of the selection of the tapes. We prepared a Supplementary Table 1 that summarizes information from the tapes to help the readers understand the tape properties. We are trying to make the CNT-PET tape commercially available very soon. I expect the first batch from RMC will hopefully come in early autumn in this year.

2) Line 165 “we used a unique slit type atmospheric pressure plasma generator (A-1000, SAKIGAKE-Semiconductor Co., Ltd., Kyoto, Japan) and a custom-made reel-to-reel motorized winder (Fig. 1i, j).” – the need for this device (and potentially its custom-design) will be an important factor in the degree to which this method is adapted, in addition to the availability of the tape. The authors need to make CAD-design plans for the winder and electrical control plans for the plasma generator available, so that other labs can readily adopt the technique.

The plasma generator A-1000 and the reel-to-reel motorized winder are manufactured goods sold by SAKIGAKE-Semiconductor Co., Ltd., Kyoto, Japan. For reader's convenience, the URL of the plasma generator is written in Materials and Methods (page 26, line 789). The CAD-design plans for the winder is shown in Supplementary figure 12.

3) Line 261: "Applicability in LM: The CNT tape has no auto-fluorescence in red or infrared light, and only very faint auto-fluorescence in green and blue. Total light transmittance is more than 90% (Supplementary Fig. 1f), therefore the tape may be used for multi-color fluorescent immunohistochemistry, as used in array tomography." This is an important point and potentially a big advantage of this tape. So the authors need to provide substantial data to characterize the light microscopic properties of this tape such as: Transmission/absorption spectrum; emission spectrum; refractive index etc. – all compared to coated Kapton. Ideally a proof-of-principle experiment using light microscopy should be shown.

We showed the optical information of the carbon coated Kapton and CNT-PET tape and capability of immunoreaction with the CNT-PET tape in Fig. 7 and summarized the results in this section (page 13, line 399~).

4) The segment about the modified staining protocol comes a bit out of the blue and does not seem absolutely essential – as in contrast to the rest of the paper, it refers more to a specific application (identification of synapses). Still, this will address the interest of a significant number of potential readers of this work, but if kept it needs a bit of a segue and some revision (lines 314-328 are too much information on already published results, so they can be shortened); also, a more systematic presentation should be adopted (e.g. in Fig 5, the TO only has to be shown to appreciate the effect of the extra heavy metals). Overall, as is, filled with similar-sounding acronyms and a somewhat unorganized back-and-forth (TO is introduced as a comparison, before it is explained; TOLA suddenly shows up combined with lead nitrate etc.), this segment is not really easy to follow. Again, a schematic of tabulated comparison of the protocols used would help and should be accompanied by a more systematical explanation of the three staining protocols (SBF, TO, TOLA) – and how TOLA derives from SBF and TO. Moreover, there is no explanation given on the fact that TOLA applies lead aspartate en bloc instead of lead nitrate in sections (which, however, can be added, see Fig 5 e,j).

We added the TO only tissue images in Fig. 9b, h which showed low contrast. We explained why we developed the TOLA method in this section. For easy understanding of each procedure, we prepared Table 1 summarizing a detailed comparison of the staining steps in the employed staining procedures.

Minor points:

1) Line 151: "The double-walled CNT tape shows superior surface conductivity and transparency over single-walled or multi-walled CNT versions." – please include data to buttress this point. Overall the paper would benefit from adding quantifications wherever possible, comparing at least the Kapton and the CNT-coated tape, such as a quantification of the major defect frequency – scratches, wrinkles, etc. Similarly, the statements about tape resilience after chemical and physical treatments could be supported by images and data derived from imaging empty films.

The double-walled CNT tape does indeed have superior surface conductivity and transparency over the other CNTs, but we learned there are many other factors, such as structural defect of CNTs, layer structure and so on. These can also determine the conductivity of the film, therefore we would rather omit this sentence.

We described the comparison of the carbon coated-Kapton tape and CNT coated-PET tape in many features and summarized some of them in Fig. 7, Supplementary Fig. 1, 8, Supplementary Table 1. We also examined the beam damage effect on the carbon coated-Kapton tape and summarized in Supplementary Fig. 4.

2) Line 176: “The plasma-treated CNT-PET tape hydrophilization lasts up to 7 months and it does not affect the surface conductivity, whereas plasma-treated Kapton tape hydrophilization lasts only a few weeks and reduces the surface conductivity because of clean-up the coated carbon particles.” This is important information and speaks to the possibility of providing ready-to-go tapes for potential users – so the data should be shown.

In addition, we found the plasma-treated effect on wrinkle generation lasted up to 13 months. We added a low magnification electron micrograph showing overall ultrathin section images without wrinkles on the tape of the 7 and 13 months after the plasma discharge treatment in Fig. 3i, j.

3) Line 181: “(87.3 % reduction of conductivity; 257 ± 4.5 ohms/square without plasma discharge treatment vs 284 ± 18.1 ohms/square with plasma discharge treatment, $n=3$ each).” At this point I got a bit confused: Which numbers were used to calculate the “87.3%”? It is a bit unclear what was compared here and how this relates to the numbers given in Line 147ff. Also, what exactly is the unit here (I do not recognize “square” as a unit)?

This unit is called sheet resistance, and is denoted ohms per square or ohms square.

https://en.wikipedia.org/wiki/Sheet_resistance

Thank you for your kind pointing out the careless mistake. The value 284 ± 18.1 should be 294 ± 18.1 ohms/square, thus the 87.3% is correct. But I just realized that we should describe it “12.6 % reduction of conductivity” (page 9, line 261).

4) Line 185ff.: The segment on Monte Carlo Simulations of track depth seems displaced, as it refers to the open reel, which the section has already bypassed. If at all necessary (the images Fig 2g-q, without a quantitative analysis – i.e. number of tracks that crosses 50nm etc. - and comparable scaling of the axes really do not help the reader much), these results should be incorporated in the tape screen section Line 111ff. Alternatively it can be shifted back - Line 214ff - to discuss the depression and its possible dependence on primary electron penetrating into the polymer (Note: This notion could be tested by imaging thicker sections and seeing whether the threshold for the formation of a depression shifts to the higher EHT values predicted by the simulations.

We described the segment on the Monte Carlo simulation in the “Open reel tape” paragraph, as the reviewer kindly advised. The new simulation with 3,000 electrons was introduced in Fig. 2 and we added a quantitative data analysis showing the number of tracks found in a 10nm bin of the depth and ES/BSE-50 value, indicating proportion of BSE only from the brain tissue section (50 nm thick) divided by all BSE including from base CNT-PET tape. We hope the new figure panels of Fig. 2 help the reader to understand the concept and significance of the Monte Carlo simulation results.

5) Line 214ff: “(Supplementary Fig. 5-8)” – as these belong together, could the results be organized into a single figure – perhaps even including SFig 4? The current design unnecessarily inflates the length of the supplement.

We enlarged the synaptic contact image in the Supplementary figures and showed in Fig. 4 and 5 the resolution difference more clearly. The

Supplementary Fig.5 and 6 with BSE detector, 7 and 8 with In-lens SE detector in the old version of the manuscript are organized into a single figure, respectively. We also add a 3D laser SCM panel showing the same place, which shows surface structure clearly with pseudo depth color. We also added figure panels showing the relation between depression depth and EHT, electron dose and EHT, depression depth and dwell time, and electron dose and dwell time for better understanding.

6) SFig 11: After how many exposure reiterations does the image quality deteriorate?
The repeated exposures were done 30 times and showed almost no image quality deterioration and shown in Supplementary Fig. 9.

7) Line 362ff: This seems like a 'loop' in the flow of the paper, back to previous topics and figures – perhaps this information can be consolidated in one place?

Thank you for pointing this out. We followed this advice and the revised paragraph was moved to right after the section for effect of the varied EHTs and dwell times on beam depression damage (page 12 line 344~).

8) Staining paragraph, line 297ff: The Hua et al., Nat Comm, 2015 paper (also cited in this article) introduced a staining protocol that separates OsO₄ penetration from ferrocyanide enhancement. This avoids unsaturated lipid depletion but allows membrane penetration. Additionally, Hua et al. established a two-step temperature protocol for UA impregnation that exploits the fact that diffusability and reactivity of UA is differentially affected. This method shows very good contrast and penetration. The authors are not stating why they did not consider these steps.

We actually use Hua's method for tissue block (about 1 mm cubic) for MSEM. I discussed this with Dr. Hua personally and he said the procedure was developed for the 1 mm cubic size block for a better penetration of the metals into deep tissue, as the reviewer mentioned. In this manuscript, we mainly use 50 µm thick brain sections with the regular single beam SEM studies, therefore we believe the conventional EM protocol with the mixed OsO₄ and potassium ferrocyanide, and a short single step of the uranyl acetate staining should work fine.

9) There is quite a bit of grammar to fix – this manuscript needs a thorough revision in this regard.

We asked an American native English speaker with a neuroscience background to revise the text. We hope the current revised manuscript is improved.

10) The figure sequence and the panel order in the figures would be reorganized to a more consistent stream to avoid back-and-forth for the reader.

We reorganized it and hope the revised form is better.

11) Overall, a broad range of sample, staining and detector types were used in this study. It was not obvious to me for all the figures what was the sample and the staining used. I would suggest to also organize this in a table or in a systematic fashion in the legends, so the reader can ensure s/he is only comparing what can be compared.

The staining method used in this study is summarized in Table 1. We would hope the reader will only compare what can be compared. We described "We embedded rat brain tissue in epoxy resin (Durcupan ACM, Sigma-Aldrich, St. Louis, U.S.A.) using a modified heavy metal staining (mHMS) histology protocol and obtained serial ultrathin sections of 50nm thickness using the

ATUMtome (RMC Boeckeler, Tucson, AZ, U.S.A.). Unless otherwise stated, we used these sections for examination." in page 5, line 116-120. We also described an information of the imaging condition in either Figure panels or legends. So, I hope it becomes a lot clearer now.

Reviewer #2 (Remarks to the Author):

The paper by Kubota et al presents a new supporting tape to be used for collecting ultrathin sections cut with the automated tape collecting ultramicrotome (ATUM). This microtome system has significant potential for the field of connectomics where very large volumes of brain tissue are needed to be imaged for the mapping of neural circuits. The automated collection of sections onto a tape, that is then imaged in the scanning electron microscope, reduces the risk of interruptions to the serial sections.

The authors show that the use of Kapton tape as suggested in the original work by Hayworth et al. is not ideal for two reasons. Firstly there is significant folding of the sections, and secondly, there is insufficient charge removal, and therefore too many imaging artefacts. This part of the paper is slightly confusing as although the authors in the current work show images of folds on sections, this was certainly never brought up in the original paper. The original paper only mentions that folding is rare. It is also noted that the original paper uses a conductive coat on the Kapton tape to reduce the charging, so I am not sure why the current paper even uses the uncoated Kapton tape as a comparison.

Thank you for your kind comments. I agree that it is not necessary to show the uncoated Kapton tape data and removed the figure panels and sentences related to the uncoated Kapton tape. About the wrinkle issue, I assume the original paper simply did not go into detail, although they probably noticed the issue. The method in Hayworth (2014) and Kasthuri (2015) was slightly different from the current method, which uses plasma discharge treated cc-Kapton tape for ultrathin section collection. For instance, Kasthuri et al. described the methods in the "Experimental procedures", that they used plasma-treated carbon-uncoated Kapton tape for section collection with the ATUMtome. After adhering the tape strips with the sections on the silicon wafers, the wafers were coated with ~ 10 nm of carbon to ensure conductivity. They actually had not described any details about the wrinkle generation in these papers, but mentioned introducing the plasma discharge treatment on the tape surface before the section collection. I believe they use the plasma treatment method to reduce the wrinkle generation of the sections for their second paper. I will try to further explain this issue and hope readers understand it better.

My other comment relates to the second aspect of the paper, which is the sample preparation and imaging conditions for this type of microscopy. They are as follows:

In figure 2, the authors look at the results obtained with magnetic tape as a support and try to explain the poor imaging results with the use of monte carlo simulation. They explain this in terms of how the electrons in the higher energy beam are interacting deeper in the section and probably with the magnetic surface. However, I am troubled as to how they can interpret this when the imaging is done with the secondary electrons, that are emerge very close to the surface, and are unlikely to traverse through significant thicknesses of resin. If electrons of any type are interacting with magnetic surface then there will be other effects. I would suggest that a magnetic surface will effect how the electrons interact with the section, and a simple Monte Carlo simulation, such as the one shown, is not sufficient to explain this, and I do not agree with the interpretation. How is it that primary electrons, so deep in the section, will cause the production of secondary electrons giving information about the magnetic surface?

Thank you for pointing out this issue. We noticed that our interpretation of the Monte Carlo simulation was not exactly correct. We modified our interpretation of the simulation. We described two occasions where the

secondary electrons are generated (page 7 line 189-191): at the primary electron incidence into the block (SE1) and upon the primary electrons back-scattering from the block surface (SE2) (Reimer, 1993). The reviewer is only considering SE1 electrons, which are generated close to the primary beam-sample interaction giving information about the surface structure. However, SE2 electrons are generated near the sample surface due to BSE electrons, which generate a large interaction volume dependent on the EHT, shown with the red line in Fig. 2: BSE electron tracks, giving information about the magnetic particles.

In Figure 3 the authors show the effect of the electron beam on the section. It is not clear why this is relevant to the paper, or how the authors are able to interpret this as a depression in the section. How was the depth of this depression measured? If it is a depression, as the authors contend, then how can they be sure this is through mass loss, and not some heating effect of the beam.

I agree with the reviewer. We revised the figure completely and showed the repeated image and depression issue in Fig. 6 and described details about the depression damage, its influence on the tile stitching and repeated imaging effect on the image quality in page 11, line 319~. I hope we described the repetitive imaging and beam damage depression issues more clearly.

Another general aspect of the paper that concerns me is that it is clearly aimed at providing the best quality of images. The authors give a description as to why they used the conditions for clearly seeing the synaptic connections. However, I notice that the images are collected with pixel dwell times that are significantly longer than were used in the original paper. Hayworth et al used very short imaging time because this is critical for being able to image large volumes in a reasonable period. It may be that synaptic connections are not imaged in optimal conditions, but the Hayworth paper explains that across serial sections the identification is straightforward. Therefore, I am concerned that the current piece of work is ignoring the fact that using their technique imaging times for a significant volume would be excessive, or even completely unfeasible.

Hayworth et al 2014 wrote "Scan speeds within a single image currently range from about 0.5 to 20 million pixels per second (MPS) using the commercially available SEMs." The dwell time range should be 50 ns - 2 μ s. In Morgan et al 2016, "Images were acquired using the in-lens secondary electron detector and a dwell time of 50 ns." They captured images with 50 ns dwell time using In-Lens detector and 2 μ s dwell time using BSE detector. We mainly captured images with 3.2 μ s dwell time using the BSE detector in the SIGMA SEM, so we believe we use comparable image capturing conditions. Morgan also wrote "High resolution images were acquired in long working distance mode using a ~3 nA current at 3.5 keV", which uses a higher current than our condition of ~0.25 nA, therefore their imaging time is quicker. I noticed that we could go slightly faster with the dwell time to get an OK image. I agree with the reviewer that the relatively slow acquisition time was a bit problematic for high throughput with large volume EM data acquisition; however, we showed the captured images with 0.1 μ sec or less using MSEM. We write "The shorter the pixel dwell time is, the worse the image quality becomes because of reduced signal. For a good quality image, imaging with a longer dwell time is always better. For high throughput, however, faster imaging is required. The balance between these two requirements depends to a large degree on the ways the tissue is prepared, and the SEM and detector are used. These parameters should be optimized for individual conditions. " in page 12 line 352~357.

There are some minor comments as well. There are instances where the authors use acronyms that are not necessary, or confusing. For example, the acronym SBF is used for the staining protocol, but this is confusing when many understand it as 'serial block face', for describing an imaging technique. The acronym EMg for electron micrograph is also strange as in many cases they could use the word, 'image'.

Thank you for the comments. I agree with you. We changed the words. The acronym SBF is changed to mHMS (modified Heavy Metal Staining). All instances of "EMG" in the text were changed to "image". I hope our manuscript is easier to read with these changes.

The paper describes the use of a number of different tapes for holding the sections, but are briefly described as not being suitable. It's not clear why these are explained in the paper if they give no advantage. It's also not clear why it's necessary to provide an image of them.

We described requirements for the ATUM tape in the last paragraph (page 4 line 98) for better understanding of the selection of the tapes. We deleted the description of the video tape, because we found it was not necessary. The image of the tapes help the reader to know the light transmittance intuitively, so we cited the figure panel appropriately. We prepared a Supplementary Table 1 that summarizes information from the tapes to help the readers understand the tape properties. We are trying to make the CNT-PET tape commercially available very soon. I expect the first batch from RMC will hopefully come early autumn in this year.

In summary, this paper gives some useful information as to the preparation and imaging of brain tissue collected onto tape for imaging with SEM. However, there are so many different variables given it's hard for the reader to understand the logic, and which are the most important. The paper presents different support films, conductive layers, stains, microscopes, voltages, dwell times, imaging modes, but does not give thorough comparisons. The paper needs significant reorganisation to convey the most important details.

Thank you for your thoughtful comments. I agree with you that our manuscript needs reorganization for our readers to more easily understand all the information contained in it. With that goal in mind, we made substantial changes and reorganized it in an effort to convey the important details and try to show comparisons between the CNT tape and cc-Kapton tape in many aspects. I hope the new version of our manuscript is easier to read and the logic is now straight forward.

Reviewer #3 (Remarks to the Author):

The paper addresses a current need for the growing number of fields that require three dimensional electron microscopy. The ATUM tape-to-SEM technology requires a stable substrate that satisfies many chemical, mechanical, and environmental criteria. To date most tape materials satisfy a subset of requirements, leaving a researcher with suboptimal performance and research results. After performing an extensive search and testing using a variety of electron microscope techniques, the authors found a carbon nanotube coated substrate that meets all the requirements of a successful tape material. The demonstration of the GABA immunolabeling helps complete the entire repertoire of how the tape may be used. The explanation of the CASINO code is a weak point in the paper and should be revised.

Line 101 Should be a 24% reduction in conductivity $\{(3367-4172)/3367\}$

Thank you for pointing out the mistake.

We calculated conductivity using this formula (page 29, line 874~880).

$$\sigma = 1 / (Rs \times t)$$

where

σ is conductivity with SI units of "siemens per meter" (S/m)

Rs is sheet resistance with SI units of "ohms per square" (Ω /sq)

t is film thickness with SI units of "meter" (m)

For cc-Kapton tape:

$$\sigma_a = 1 / (4172 \times 0.00005) = 4.7939 \text{ S/m}, \sigma_b = 1 / (3367 \times 0.00005) = 5.94 \text{ S/m}$$

Reduction in conductivity = $1 - \sigma_a / \sigma_b = 1 - 4.7939 / 5.94 = 0.1929$: The reduction is 19.3% and shown in page 5, line 136.

Where σ_a is conductivity after the plasma treatment, σ_b is conductivity before the plasma treatment.

We also calculated reduction in conductivity for the CNT tape in the same way and the reduction was 12.6% (page 9, line 261).

Line 304 Striped pattern not obvious in supplemental 3c,d

We indicated the striped pattern with arrows in Fig. 1c, d. We also wrote a possible reason for the generation of the striped pattern in ITO background noise, which we believe is due to the ITO layer cracking at the very small angle of the ATUMtome tape guide tip end before collecting serial thin sections (page 6, line 162~).

Line 185 through line 200: This paragraph is poorly executed. The concepts of interaction volume, range, straggle and escaped depth should be discussed in the context of the section thickness and underlying substrate. If this is not done, the paragraph should be removed and figure 2 reworked.

We revised the paragraph more concisely and the Monte Carlo simulation analyses were done using 3000 electrons. We reworked the figure 2. We carefully reconsidered the interpretation of the simulation and discussed it in the context of the section thickness (50 nm) and underlying substrate, CNT or carbon (page 7, line 179~). I hope this satisfies the potential readers.

Lines 210 and 211: Can the effects of mass loss be expounded upon? How will the depressions affect downstream efforts such as the stitching of mosaics and the 3D alignment of the serial sections.

We described details for the depression damage in page 10 line 281~. We showed that the stitching of mosaics could be done without any problems using whole mouse brain tissue section and the MSEM in the Fig. 8. We imaged a large area including 4 imaged domains and checked the image at the border between the imaged area and the non-imaged area shown in Fig. 6 and described details in page 11, line 319~. We found the depression artefact would not have any effect on the image and we believe the downstream efforts, such as the stitching of mosaics and the 3D alignment of the serial sections, will not have any associated problems. This was because the focus depth of the detectors was sufficiently large, such that the image from the depression area could still be clearly focused.

Line 223: Linking the cause of the depression to additional crosslinking of the overcoat layer is made; however, earlier, the process of mass loss was used to describe the reason for the depression. Please better explain what is causing the depression. Given the importance of the CNT/overcoat layer, the overcoat layer polymer should be identified along with the thickness of the layer.

I am sorry that we forgot to change “mass loss” to “further crosslinking” in page 11, line 282 in the initial submission. We described a possible reason for the cross-linkage of the overcoat layer in page 10, line 284-287. We measured the thickness of the CNT/overcoat layer with a laser confocal microscope and estimated it is about 2 μm and shown in page 8, line 225.

There does not seem to be a distinction made between using a SE2 detector and an in-lens detector (Explain why figure 3c does not look like figure 2 u)

Thank you for pointing out this issue. We revised “SE2” detector as “side-mounted Everhart-Thornley (sET)”, which captures surface structure including bumps and images of the tissue, and the “SE” detector as “In-lens SE detector”, which captures images of the tissue and some surface information.

Figure 1: The total tape thickness is not defined. Figure 1i is does emphasizes the control panel, but not the reel-to-reel or plasma system.

The layer thickness of the tape is defined in Figure legends and in the text (page 8, line 225-226). We added the reel-to-reel winder and plasma torch photos in this figure. The figure was moved to Fig. 3.

Figure 3 SMST is not defined.

We apologize this mistake. It should be SBF, but now mHMS (modified heavy metal staining) and the figure, itself, was moved to Fig. 6.

Figure 4 The images in this figure4 d and e, should be denoted as single-beam backscattered images, as distinct from the mSEM secondary electron images

I noted it in the legend and the figure. The figure is now Fig. 8.

Figure 5: The molecular formula for lead nitrate(II) is $\text{Pb}(\text{NO}_3)_2$. Using ‘In’ is not very informative. If the abbreviation is made for lack of space, maybe use PbN.

I changed the abbreviation to Pbc, because we used a lead citrate staining method, which uses lead nitrate (II). We added a description of how to make the lead citrate solution in page 23, line 700-703 in Materials and Methods.

Supplementary Figure 3 does not contain an example image of carbon-coated Kapton tape.

The supplementary Figure 3 is moved to Fig. 1 and example images of carbon-coated Kapton tape were added in Fig. 1a, b.

Supplementary Figure 4: Can you comment on the physical mechanism of the depression?

We add a comment in Page 10, line 284-287.

We believe this depression may be due to further crosslinking of the polymer in the overcoat layer of the CNT tape caused by the heat generated from the SEM beam.

Supplementary Figure 5: I am not sure that I understand the point of this figure. To begin we are not told that the working distance was optimized to collect an image with the largest signal to noise ratio. Second, the images shown are more a function of the backscattered electron detector properties, rather than a collective property of the other imaging parameters such as, EHT or dwell time. For instance the energy response function of the BS detector will determine the SNR for a given EHT, while the bandwidth of the detector will determine the amount of noise in the image for a given dwell time. So saying that 4 and 5 kV are acceptable while 3 and 6 kV are unacceptable is only accurate for this specific backscatter detector and not a general result.

We described the optimized working distance for detectors in the figure legend. We absolutely agree with your second opinion. Therefore, we showed a variety of EM systems. Mainly we showed results collected with our SIGMA and detectors (BSE, In-lens SE, sET) and also showed images obtained with the other SEM (QuantaFEG 200, GeminiSEM 300 and Multi SEM). We would like to address the optimized conditions should be evaluated for individual SEM systems to get the best images as your advice in page 12, line 351-357: "It is essential for individual users to investigate the optimized EHT for imaging. The shorter the pixel dwell time is, the worse the image quality becomes because of reduced signal. For a good quality image, imaging with a longer dwell time is always better. For high throughput, however, faster imaging is required. The balance between these two requirements depends to a large degree on the ways the tissue is prepared, and the SEM and detector are used. These parameters should be optimized for individual conditions."

Supplementary Figure 6: The dose rate (#electrons/unit area/time) and the total electron dose (#electrons per unit area) for each condition should be stated. Though the image suggests that increasing the EHT increases the depression depth, the evidence of the depression depth not changing with dwell time is lacking. If the authors want to make this claim, the depression depths should be measured.

In the revised version, this figure is Supplementary figure 5. The dose rate and the total electron dose were summarized in panels e, g. We analyzed the depression depths statistically and found a significant difference in EHTs (panel f), but not in dwell times (panel d).

Supplementary Figure 7: The dose rate (#electrons/unit area/time) and the total electron dose (#electrons per unit area) for each condition should be stated. The type of SE detector should be described (Everhart-Thornley or in-lens). It is unlikely that the darker center regions result from surface depressions. More likely is that these are focusing squares and the darkening is the result of a thin layer of carbon (adventitious) build up. In addition, a contrast reversal is apparent in the 1kV images is seen. This should be acknowledged and commented upon. Again, there is no comparison to the carbon-coated Kapton.

See the response to the previous question.

Supplementary Figure 8: The SE images now clearly show that the scan depressions

not only are a function of accelerating voltage, but also a function of dwell time. This observation is made possible by the surface sensitivity of secondary electrons. This means that the conclusion in Supplementary Figure 6 is probably incorrect.

I agree with your opinion. The depression depth is significantly correlated with the EHT (Supplementary Fig. 6f). It seems the depression depth may show a very small correlation with the dwell time, but only for images captured with In-lens SE, as shown in Supplementary Fig. 6c, d, but not for images captured with BSE. However, we could not find a significant difference in the different dwell times (Supplementary Fig. 6d).

Supplementary Figure 9: Imaging parameters for all images should be described. In general, the images in this figure have poor signal to noise ratio and narrow histograms. Higher quality images at higher magnifications (structures of interest should be easily seen) should be acquired.

We showed the imaging parameters for all images and modified the figure. We adjusted the images to be as good as possible. I hope they look better.

Reviewers' comments:

Reviewer #1 (Remarks to the Author):

This is a resubmitted manuscript by Kubota and colleagues that describes the search for and characterization of a novel support tape for automated tape-collecting ultramicrotomy (ATUM); further, tape conditioning procedures and adaptations of staining protocols are presented to optimize the use of this new support material.

The authors have revised their original submission and have provided answers to most questions that the other reviewers and I raised; they have also revised and improved the figures. My feeling always was that from a scientific point of view, this is a valuable contribution and should be published at some point. Indeed, my specific scientific points have been satisfactorily addressed, in several cases with new data (e.g. on the utility of the new tape in fluorescence microscopy). However, I also had criticism and questions related to the way the results were presented, which made the information harder to access than necessary and gave the manuscript a meandering and unnecessarily lengthy feel. While the authors have addressed many of the specific points here and have improved some aspects, I still found the manuscript a tedious read and unnecessarily opaque. I am wondering whether this paper would not benefit from being cut substantially, focused on the main point (the new tape that works, at the expense of all the information on variants that did not work so well; which at best could go into a supplementary methods part), and strive for a concise presentation. Just a few, somewhat random examples - Figure 2; the figure's format makes the relevant point hard to grasp, because there are several rescales in the trajectory graphs (which no one will read in detail anyway), accompanied by rotated histograms with inserted tables (which will be illegible in the final paper). I could easily see a presentation of this information that at best would occupy two panels in a multi-panel figure. Similarly, in Figure 4 and 5 we are still just presented with a matrix of images taken at various conditions - I cannot tell what is "optimum" here; plus we are told (in discussion) that the specific settings depend on the instrument anyway, so where is the general point? Should not rather relevant parameters (contrast, signal-to-noise etc.) be extracted and presented in a way where the specific value on an axis might vary between instruments, but the general (quantitative) trend is actual information? Same goes for Figure 1, which in its caption contains the central claim of the paper ("CNT-PET tape allowed for the highest quality images of ultrathin sections"), but does not allow the reader to appreciate this claim. As a final example, the plasma discharge device presented in Figure 3 - we now not only see the photos we had seen originally, but a few more and have a technical drawing in the supplement - this is helpful, but it is still not unambiguous how this device exactly works, as e.g. I do not know what the three wheels are for, how the motorized wheel grabs the tape etc, as there is little labeling and explanation; I can guess these things, and it might not be important, as it is a pretty trivial device that just pulls a tape pass a slit, but then why spend so much room on it, including a movie that shows little (but has piano music, perhaps to honor Kristen Harris?).

So my bottom line is - I believe this paper can be published, but I am not sure it is optimized; it boils down to an editorial decision of how much to invest here. Most interested readers, who really need the information will get it, but the presentation will not add readers beyond that narrow circle. So I would either recommend another overhaul to optimize the paper (which will need some editorial input and can then be judged editorially), or perhaps a more specialized journal (such a one from the Frontiers series).

Reviewer #2 (Remarks to the Author):

This paper is improved from the initial submission, but only marginally. I still feel that although some useful information is presented, it's far too long-winded, and provides a huge amount of extraneous details that are not needed.

As it currently stands it's hard for the reader to fully understand the most important points. The focus of this paper is a new type of tape for collecting serial sections generated with the automated tape collecting ultramicrotome. However, the paper's results section begins by evaluating other types of tapes first and all these results are presented in the supplementary section, and it distracts from the final message. All references to figures in the first couple of pages are supplementary and therefore would appear to be unimportant for the main thrust. However, if these tests are to show that these other tapes are not useful for the purposes required, why are they also shown in the first main two figures?

I would suggest that the paper begins by stating that the purpose of the paper is evaluate the usefulness of the CNT-PET tape, and comparisons have been made with the commonly used alternative, Kapton and carbon coated Kapton. I don't understand why the other types of tape are even mentioned as they are clearly not useful, and certainly not being used by other labs. If the authors have a good reason for showing the data, then they must be moved to the supplementary. As it stands, all this other data clouds some interesting results which should be presented more succinctly. I would also suggest that Figure 1 and 2 are removed.

The second part of the paper is a modified version of a heavy metal stain, and this should be presented as the second main part, and in comparison with the more commonly used versions. This part should not be presented in the final figure either.

For the comparison of imaging quality with different electron beam parameters, it is not clear to me as to why different synapses are being shown in each panel. Would it not be better to show the exact same region, with only the beam properties changing so that the reader can easily see the difference? I would also suggest the number of parameters shown is reduced.

I am also unclear as to what the purpose of Figure 8. If the tape works with single beam SEM, why would multi-beam be any different? The paper is about the supporting medium and a comparison with others. It's confusing to add in another type of microscope at the end of the paper.

Reviewer #3 (Remarks to the Author):

Kubota et al.'s revisions have significantly increased the readability and impact of the paper. The paper's topic provides a timely solution to an important technical problem relevant to the serial electron microscopy field.

Below are some minor points that should be addressed.

Figure 1: Since the cc-Kapton is the most widely used tape, the images from the CNT and the cc_kapton should be placed side-by-side: CNT-PET tape allowed for the highest quality images of ultrathin sections: It is difficult to tell from the images presented. A better image comparison should be made, or the sentence rewritten with a "softer" comparison.

Cacodylate is misspelled

The CNT tape is uniform? The CNT tape is electrically uniform, but not uniform in structure.

Can the authors comment on how will the CNT structure might affect stitching and aligning of images? Has this experiment been performed? If not, the authors should acknowledge that the CNT structure may or may not complicate downstream alignment processes.

The authors should comment on the commercial availability of the CNT tape.

Reviewers' comments:

Reviewer #1 (Remarks to the Author):

This is a resubmitted manuscript by Kubota and colleagues that describes the search for and characterization of a novel support tape for automated tape-collecting ultramicrotomy (ATUM); further, tape conditioning procedures and adaptations of staining protocols are presented to optimize the use of this new support material.

The authors have revised their original submission and have provided answers to most questions that the other reviewers and I raised; they have also revised and improved the figures. My feeling always was that from a scientific point of view, this is a valuable contribution and should be published at some point. Indeed, my specific scientific points have been satisfactorily addressed, in several cases with new data (e.g. on the utility of the new tape in fluorescence microscopy).

 Thank you for your kind words.

However, I also had criticism and questions related to the way the results were presented, which made the information harder to access than necessary and gave the manuscript a meandering and unnecessarily lengthy feel. While the authors have addressed many of the specific points here and have improved some aspects, I still found the manuscript a tedious read and unnecessarily opaque. I am wondering whether this paper would not benefit from being cut substantially, focused on the main point (the new tape that works, at the expense of all the information on variants that did not work so well; which at best could go into a supplementary methods part), and strive for a concise presentation.

 Thank you for your kind comments. I agree with you. I understand the manuscript should concentrate on introducing the fact that that CNT tape is a good alternative tape for the ATUM-SEM system. We moved the text of 6 pages and some more, including the paragraphs in the first three pages: "Drawbacks of cc-Kapton tape", "Examinations of potential tapes: Copper foil, germanium-coated and ITO-coated PET tape", "Open reel tape" and some of the "Properties of the CNT-coated PET tape", to supplementary

information and the main text was re-organized. The main text now starts with a brief description of the cc-Kapton tape and is followed by the CNT tape in the first page of the Results. I hope that makes the manuscript more straightforward and easier to understand

Just a few, somewhat random examples - Figure 2; the figure's format makes the relevant point hard to grasp, because there are several rescales in the trajectory graphs (which no one will read in detail anyway), accompanied by rotated histograms with inserted tables (which will be illegible in the final paper). I could easily see a presentation of this information that at best would occupy two panels in a multi-panel figure.

 We moved the Figure 2 to Supplementary information and modified it to make a new Supplementary figure 8. All the trajectory graphs are in the same scale shown in panel b and peak plot distribution graphs for BSEs and primary electrons were added in panel c. I hope it is now easier to see for the readers.

Similarly, in Figure 4 and 5 we are still just presented with a matrix of images taken at various conditions - I cannot tell what is "optimum" here; plus we are told (in discussion) that the specific settings depend on the instrument anyway, so where is the general point? Should not rather relevant parameters (contrast, signal-to-noise etc.) be extracted and presented in a way where the specific value on an axis might vary between instruments, but the general (quantitative) trend is actual information?

 We replaced the Figure 4 and 5 with the Figure 7. The same region was repeatedly captured using various acceleration voltages to see whether there was any image quality difference found between the images. We believe some difference was visually observed. We extracted image quality quantitatively using 'Michelson contrast' and 'Contrast-to-noise ratio' from an intensity histogram as the reviewer #1 kindly commented regarding this issue and we found images with 3/4 keV using a BSD, and with 1.5/2 keV using an In-lens SE provided good results. This quantitative image quality analysis coincided with our visual impression.

I hope I can address "The higher the acceleration voltage is, the greater the background noise from the CNT tape becomes, and, as a result, image contrast deteriorates. The lower the acceleration voltage is, the lower the image contrast becomes, due to reduced signal." (page 11 line 332 - 335) as the general comments.

Same goes for Figure 1, which in its caption contains the central claim of the paper ("CNT-PET tape allowed for the highest quality images of ultrathin sections"), but does not allow the reader to appreciate this claim.

 I regret that the Figure 1 caption was incorrect. We modified the description to "CNT-PET tape allowed the comparable images of ultrathin sections with cc-Kapton tape." as we wrote in the main text. It is now Figure 2.

As a final example, the plasma discharge device presented in Figure 3 - we now not only see the photos we had seen originally, but a few more and have a technical drawing in the supplement - this is helpful, but it is still not unambiguous how this device exactly works, as e.g. I do not know what the three wheels are for, how the motorized wheel grabs the tape etc, as there is little labeling and explanation; I can guess these things, and it might not be important, as it is a pretty trivial device that just pulls a tape pass a slit, but then why spend so much room on it, including a movie that shows little (but has piano music, perhaps to honor Kristen Harris?).

 The right-side panels in Figure 3 go into supplementary figure 1. We added a precise description of the custom made reel-to-reel winder in Methods page 21, lines 620-629 and some words were described on the Supplementary Figure 1g. The supplementary video 1 was removed.

So my bottom line is - I believe this paper can be published, but I am not sure it is optimized; it boils down to an editorial decision of how much to invest here. Most interested readers, who really need the information will get it, but the presentation will not add readers beyond that narrow circle. So I would either recommend another overhaul to optimize the paper (which will need some editorial input and

can then be judged editorially), or perhaps a more specialized journal (such a one from the Frontiers series).

 We really hope the revised version is improved.

Reviewer #2 (Remarks to the Author):

This paper is improved from the initial submission, but only marginally. I still feel that although some useful information is presented, it's far too long-winded, and provides a huge amount of extraneous details that are not needed.

As it currently stands it's hard for the reader to fully understand the most important points. The focus of this paper is a new type of tape for collecting serial sections generated with the automated tape collecting ultramicrotome. However, the paper's results section begins by evaluating other types of tapes first and all these results are presented in the supplementary section, and it distracts from the final message. All references to figures in the first couple of pages are supplementary and therefore would appear to be unimportant for the main thrust. However, if these tests are to show that these other tapes are not useful for the purposes required, why are they also shown in the first main two figures?

 Thank you for your kind comments. I agree with you. I understand the manuscript should concentrate on introducing the fact that that CNT tape is a good alternative tape for the ATUM-SEM system. We moved the text of 6 pages and some more, including the paragraphs in the first three pages: "Drawbacks of cc-Kapton tape", "Examinations of potential tapes: Copper foil, germanium-coated and ITO-coated PET tape", "Open reel tape" and some of the "Properties of the CNT-coated PET tape", to supplementary information and the main text was re-organized. The main text now starts with a brief description of the cc-Kapton tape and is followed by the CNT tape in the first page of the Results. I hope that makes the manuscript more straightforward and easier to understand

I would suggest that the paper begins by stating that the purpose of the paper is evaluate the usefulness of the CNT-PET tape, and comparisons have been made with the commonly used alternative, Kapton and carbon coated Kapton. I don't understand why the other types of tape are even mentioned as they are clearly

not useful, and certainly not being used by other labs. If the authors have a good reason for showing the data, then they must be moved to the supplementary. As it stands, all this other data clouds some interesting results which should be presented more succinctly. I would also suggest that Figure 1 and 2 are removed.

 We now reorganized the main text. We described comparisons of the CNT tape and cc-Kapton tape and introduce how is the CNT tape superior to cc-Kapton tape in many features.

I just would like to introduce how we struggled with using the other tapes in the beginning and hopefully the potential readers can avoid having to make similar mistakes. This is the main reason to introduce the different tapes. Indeed, we know a several people have been extensively looking for alternatives to cc Kapton tape. What we find to be a pity in science is that usually only success stories are published. Therefore, we believe it is worthwhile to report in full detail all the things we have tried and that did not work. I understand what the reviewer #2 said very well, so I put all the paragraphs to Supplementary information, and found the main text is much easier to read. I do appreciate for your kind comments.

The second part of the paper is a modified version of a heavy metal stain, and this should be presented as the second main part, and in comparison with the more commonly used versions. This part should not be presented in the final figure either.

 Thank you for pointing out this. We followed to your advice and found our manuscript improved. The images of tissue sections made with different histological procedures on the cc-Kapton tape are also added in Figure 6 m-r. The Figure 6 suggests that the SEM images with the CNT tape and cc-Kapton tape, and TEM images are comparable.

For the comparison of imaging quality with different electron beam parameters, it is not clear to me as to why different synapses are being shown in each panel. Would it not be better to show the exact same region, with only the beam properties changing so that the reader can easily see the difference? I would also suggest the number of parameters shown is reduced.

 All right. That is a good idea. We replaced the Figures 4 and 5 with new Figure 7. We captured the same region with one variable parameter, accelerating voltage, to compare the image quality. We also added new analysis 'Michelson contrast' with intensity histogram to understand the image quality quantitatively.

I am also unclear as to what the purpose of Figure 8. If the tape works with single beam SEM, why would multi-beam be any different? The paper is about the supporting medium and a comparison with others. It's confusing to add in another type of microscope at the end of the paper.

 I have a good reason for this and add this paragraph in Discussion, page 14, lines 422 - 431. I hope this sufficiently addresses the issue.

"The MultiSEM has a total current of 61 channels, 570 pA per beam. In single beam set-ups we use 200 pA to 3.2 nA. In the MultiSEM this is 10 to 175 times more current. In addition to that, the stage bias of 28.5 keV is also much higher than in a single beam microscope. Both beam current and stage bias require absolutely good conductive properties of the tape. Practically we observed cases where commercially available ccKapton tape works well in single beam SEM but charged a lot in the MultiSEM (the cc-Kapton tape with surface resistance of 6,530 Mohms/square). From this point of view, we believe the CNT tape, which surface resistance is about 240 ohms/square, provides a good imaging condition for the MultiSEM, which may not be always promised with the cc-Kapton tape."

Reviewer #3 (Remarks to the Author):

Kubota et al.'s revisions have significantly increased the readability and impact of the paper. The paper's topic provides a timely solution to an important technical problem relevant to the serial electron microscopy field.

 I appreciate for your kind words very much.

Below are some minor points that should be addressed.

Figure 1: Since the cc-Kapton is the most widely used tape, the images from the CNT and the cc kapton should be placed side-by-side: CNT-PET tape allowed for

the highest quality images of ultrathin sections: It is difficult to tell from the images presented. A better image comparison should be made, or the sentence rewritten with a "softer" comparison.

 Thank you for your kind comments. Now the new Figure 2 showed this. I regret that the Figure 1 caption was incorrect. We modified the description to "CNT-PET tape allowed the comparable images of ultrathin sections with cc-Kapton tape." as we wrote in the main text.

Cacodylate is misspelled

 Thank you. I corrected them in Table 1.

The CNT tape is uniform? The CNT tape is electrically uniform, but not uniform in structure.

 You are correct. I corrected the descriptions to "uniform surface resistance" and added a new analysis showing the "uniform surface resistance" in Figure 1b.

Can the authors comment on how will the CNT structure might affect stitching and aligning of images? Has this experiment been performed? If not, the authors should acknowledge that the CNT structure may or may not complicate downstream alignment processes.

 We showed an example in Fig. 8b and c in our previous manuscript version. They are stitched tile images. The stitching was quite well done. We also added a new figure 4 showing the stitching the tiles was quite easily and smoothly done. The challenge could be aligning consecutive slices if the presence of artifacts is very high. We did not see the presence of the high artifacts from the tape and we had succeeded the aligning of serial 300 sections in the other research project, so it should not be a big issue.

The authors should comment on the commercial availability of the CNT tape.

 We provided this in the supplementary table 1 as "commercially available by RMC in 2018". The pilot tape had been distributed just recently

to some ATUM users to get their first feedback and hopefully the CNT tape will be released very soon commercially.

REVIEWERS' COMMENTS:

Reviewer #2 (Remarks to the Author):

The paper by Yoshiyuki Kubota is much improved from the previous version. The manuscript is now more focussed on the central issue, which is an improved version of the tape used to collect serial sections from scanning electron microscopy. I feel this paper gives a good overview of these methodological refinements and will be well-received.

There are, however, a couple of minor points that I would like to raise. The first is to do with the author's repeated statements about the usefulness of their improvements for connectomics and large-scale imaging efforts. This is a little disingenuous. I fully appreciate that this new tape and staining protocol will provide high-quality images, as this is what the authors have concentrated their efforts on, but the parameters are not useful for imaging a large volume, with a single beam. The pixel sizes they have tested are too small and the dwell time too long. Therefore, I feel the authors are perhaps ignoring the fact that the current heavy metal protocols may be staining a little too densely for a very good reason - so that the electron beam can scan very quickly. The quality may not be as good, but its still adequate for detecting synapses and neurites in very large volumes, and in a reasonable time frame.

The second minor point is that I am not sure as to why Figure 4 is included. It does not appear to be that useful. I would suggest it is removed, but the result could be mentioned in the text.

Reviewers' comments:

Reviewer #2 (Remarks to the Author):

The paper by Yoshiyuki Kubota is much improved from the previous version. The manuscript is now more focussed on the central issue, which is an improved version of the tape used to collect serial sections from scanning electron microscopy. I feel this paper gives a good overview of these methodological refinements and will be well-received.

There are, however, a couple of minor points that I would like to raise. The first is to do with the author's repeated statements about the usefulness of their improvements for connectomics and large-scale imaging efforts. This is a little disingenuous. I fully appreciate that this new tape and staining protocol will provide high-quality images, as this is what the authors have concentrated their efforts on, but the parameters are not useful for imaging a large volume, with a single beam. The pixel sizes they have tested are too small and the dwell time too long. Therefore, I feel the authors are perhaps ignoring the fact that the current heavy metal protocols may be staining a little too densely for a very good reason - so that the electron beam can scan very quickly. The quality may not be as good, but its still adequate for detecting synapses and neurites in very large volumes, and in a reasonable time frame.

 Thank you for your kind comments. I agree with you. Regarding to this issue, I added sentences in the Discussion, page 13, lines 379 -386. I hope this sufficiently addresses the issue. "The stronger metal staining protocol, mHMS stains the tissue too densely, resulting in images that make the identification of inhibitory synapses difficult, but with contrast adequate for detecting most excitatory synapses. An advantage of heavy metal staining is that the electron beam can scan samples quickly, that is, in a few hundred nanoseconds pixel¹, using an In-lens SE detector, and can acquire very large volumes within a reasonable time. In contrast, images from tissue sections processed with the TOLA protocol require more time to acquire, but these samples have well-preserved ultrastructure suitable for the analysis of synapses. "

The second minor point is that I am not sure as to why Figure 4 is included. It does not appear to be that useful. I would suggest it is removed, but the result could be mentioned in the text.

 I agree that it is probably redundant. I moved Figure 4 to the Supplementary information as Supplementary Figure 8 and added just one sentence in the main text, page 7, lines 181 -182. "We then verified that the depression damage caused by the electron beam did not cause subsequent stitching problems (Supplementary Fig. 8). "